# DIRECTIONAL INFLUENCE FUNCTION: ESTIMATING TRAINING DATA INFLUENCE IN CONSTRAINED LEARNING

## ABSTRACT

Constrained learning has been increasingly applied to various domains to ensure explicit feasibility requirements due to fairness, safety, robustness, regularization, and physics or logic constraints. Understanding how training samples influence the solution (e.g., learned parameters) of constrained learning is crucial for interpretability and robustness. The classical influence function (IF) may becomes unreliable in constrained settings: data perturbations can reshape both the objective and the feasible region, leading to estimates that violate feasibility. In response, we propose the Directional Influence Function (DIF), a new estimator that explicitly incorporates the constraints into influence estimation. DIF formulates the optimality conditions of constrained learning as a variational inequality (VI) and analyzes how perturbing training data affects this VI. We validate DIF in constrained linear regression and demonstrate that it recovers leave-one-out retraining results, whereas IF and penalty-based IF exhibit significant bias. We further apply DIF to fairness-constrained CNNs, where DIF accurately predicts test loss changes under data removal and aligns closely with actual retraining. Our results establish DIF as an efficient and reliable tool for data attribution in constrained learning.

## 1 INTRODUCTION

Understanding how individual training samples influence model predictions, also known as data attribution (Pruthi et al., 2020; Feldman & Zhang, 2020; Lin et al., 2024), is fundamental for interpreting model behavior, model correction, and data error debugging. Although retraining the model after removing a sample and comparing the resulting change in model solution (learned parameters) provides ground-truth data influence, this approach is computationally expensive for modern deep learning models. A widely adopted alternative is to approximate sample influence by studying how the solution responds to small data perturbations. As a representative method, the influence function (IF), originally developed in robust statistics (Hampel, 1974) and later adapted to machine learning (Koh & Liang, 2017; Feldman & Zhang, 2020; Zhang et al., 2024), instantiates this idea by differentiating the solution with respect to (w.r.t.) small data perturbations. While these influence estimators work for unconstrained learning problems, they often fail when applied to constrained learning. The latter refers to learning tasks constrained by domain knowledge, such as physics-informed neural networks (PINNs), or those constrained by embedding requirements, including fairness, safety, and robustness (Hounie et al., 2023; Li et al., 2023; García & Fernández, 2015; Xia et al., 2025). The empirical formulation of constrained learning is (Chamon & Ribeiro, 2020):

$$\bar{\theta} := \arg\min_{\theta \in \mathbb{R}^d} \quad \frac{1}{N_0} \sum_{i=1}^{N_0} \ell_0\left(z_i^{(0)}, \theta\right)$$

$$\text{s.t.} \quad \frac{1}{N_j} \sum_{i=1}^{N_j} \ell_j\left(z_i^{(j)}, \theta\right) \leq \tau_j, \quad j = 1, \ldots, m, \tag{1}$$

where $\theta \in \mathbb{R}^d$ denotes the model parameters to be optimized, and $\bar{\theta}$ is the optimal solution. The function $\ell_0\left(z_i^{(0)}, \theta\right)$ represents the primary loss evaluated on the training data point $z_i^{(0)}$. Each

constraint $j \in \{1, \ldots, m\}$ is represented by an auxiliary loss function $\ell_j\left(z_i^{(j)}, \theta\right)$ evaluated over a separate dataset $\left\{z_i^{(j)}\right\}_{i=1}^{N_j}$. The constant $\tau_j$ specifies the upper bound allowed for the $j$th constraint. The influence estimators aim to quantify the change in solution, denoted by $\Delta\theta$, when specific training data points participating in the objective function or constraints (e.g., $z_1^{(1)}$) are removed. Current IF-based approaches fail in constrained learning for two primary reasons. First, constrained problems (1) require that the perturbed solution, $\bar{\theta} + \Delta\theta$, remains within the feasible region, while IF methods ignore the constraints, not to mention that the feasible region itself may also be altered by data perturbation. Existing IF cannot guarantee that the estimated solution change is feasible. Second, IF primarily relies on the gradient of the optimal solution $\bar{\theta}$ w.r.t. data perturbations. In constrained learning, the solution can be only directionally differentiable. The full derivatives do not necessarily exist, rendering IF estimators invalid. This issue arises because the solution updates must adhere to the feasible region when constraints are active, often requiring projections to maintain feasibility. Additionally, data removal can change the active constraint set, leading to sudden shifts in the optimal solution. See the problem (3) in Section 2.1 for a concrete example.

To overcome the above limitations, this paper introduces the *Directional Influence Function* (DIF), an influence function designed for constrained learning, to estimate the impact of training points in either the loss function or the constraints on the model solution. DIF estimates the change in model solution through a directional derivative approach, aiming to address the following question:

> *How does the solution of constrained learning change when data points are removed from either the objective loss function or the constraints?*

The main contributions of the paper are: (1) We formalize data attribution for constrained learning by casting the optimality conditions as a variational inequality (VI) and performing local sensitivity analysis of this VI; (2) DIF quantifies the effect of data perturbations on model solutions via directional derivatives, thereby addressing the non-smoothness of solution changes induced by constraints; and (3) DIF can be efficiently computed by solving a quadratic program (QP); when all constraints are inactive (i.e., their KKT multipliers are zero), DIF reduces to the classical IF.

The remainder of this paper is organized as follows. Section 2 formally defines our problem and illustrates the failure of IF in a constrained learning setting through a linear regression example. Section 3 introduces the proposed DIF. Section 4 analyzes the impact of data perturbations on the VI-formulated optimality conditions, introduces a QP approach to compute DIF, and presents our main theoretical results. Section 5 evaluates DIF on severl learning tasks, followed by concluding remarks in Section 6. Throughout this paper, we assume that all functions involved are twice continuously differentiable ($C^2$). A detailed comparison with prior work is also presented in Appendix A.

## 2 PROBLEM FORMULATION

Let $Z^{(0)} = \left\{z_i^{(0)}\right\}_{i=1}^{N_0}$ be the dataset for the objective function and $Z^{(j)} = \left\{z_i^{(j)}\right\}_{i=1}^{N_j}$ the dataset for the $j$-th constraint. We begin by examining how the solution $\bar{\theta}$ changes when a selected subset $Z^r \subset \bigcup_{j=0}^{m} Z^{(j)}$ is removed. This removal is modeled via a perturbation formulation, where each point in $Z^r$ is assigned a small weight $\varepsilon_k : \varepsilon_0$ for points contributing to the objective, and $\varepsilon_j, j \in \{1, \ldots, m\}$ for the points contributing to the $j$-th constraint. This formulation, referred to as *Perturbed Constrained Learning* (PCL), is defined as follows[1]:

$$
\min_{\theta} \quad \frac{1}{N_0} \sum_{i=1}^{N_0} \ell_0\left(z_i^{(0)}, \theta\right) + \varepsilon_0 \sum_{z_i^{(0)} \in Z^r} \ell_0\left(z_i^{(0)}, \theta\right)
$$

$$
\text{s.t.} \quad \frac{1}{N_j} \sum_{i=1}^{N_j} \ell_j\left(z_i^{(j)}, \theta\right) + \varepsilon_j \sum_{z_i^{(j)} \in Z^r} \ell_j\left(z_i^{(j)}, \theta\right) \leq \tau_j, j = 1, \ldots, m. \tag{2}
$$

---

[1]To simplify our discussion, this constrained learning formulation does not explicitly include equality constraints, which can be considered straightforwardly.

Let $\varepsilon = [\varepsilon_0, \ldots, \varepsilon_m]$ denote the perturbation vector. When $\varepsilon = \mathbf{0}$, problem (2) reduces to the problem (1). Shifting $\varepsilon$ from $\bar{\varepsilon} = [0, 0, \ldots, 0]$ to $\hat{\varepsilon} = \left[-\frac{1}{N_0}, -\frac{1}{N_1}, \ldots, -\frac{1}{N_m}\right]$ corresponds to removing $Z^r$ from the constrained learning (1). Treat $\theta$ as a function of $\varepsilon$, denoted by $\theta(\varepsilon)$, and define the solution change as $\Delta\theta = \theta(\hat{\varepsilon}) - \theta(\bar{\varepsilon})$. By definition, $\theta(\bar{\varepsilon}) = \bar{\theta}$ is the solution to problem (1).

## 2.1 Failure of IF in Constrained Learning

**Toy example**. We use a $\ell_2$-constrained least-squares example to demonstrate the failure of IF in constrained learning, as shown below. Here $\theta = [\theta_1, \theta_2]$ are the model parameters.

$$\min_{\theta} \quad \frac{1}{3}(\theta^\top x_1 - y_1)^2 + \frac{1}{3}(\theta^\top x_2 - y_2)^2 + \frac{1}{3}(\theta^\top x_3 - y_3)^2 + \varepsilon \cdot (\theta^\top x_2 - y_2)^2 \tag{3}$$
$$\text{s.t.} \quad \|\theta\|_2^2 \leq 0.5.$$

The data are as follows: $[x_1, x_2, x_3] = [(1, 0); (1, 0); (0, 1)]$, $[y_1, y_2, y_3] = (1, 0, 1/2)$. The explicit solution of this problem is:

$$\theta(\varepsilon) = \text{Proj}_{\|\cdot\|_2^2 \leq 0.5}\left[\left(\frac{1}{2 + 3\varepsilon}, 0.5\right)\right] \tag{4}$$

Removing $(x_2, y_2)$ is equivalent to shifting $\varepsilon$ from 0 to $-\frac{1}{3}$. IF defines the impact caused by data removal using the derivative of $\theta(\varepsilon)$:

$$IF(z_2) = \left.\frac{d\theta(\varepsilon)}{d\varepsilon}\right|_{\varepsilon=0} = -H_{\bar{\theta}}^{-1}\nabla_\theta \ell_0(x_2, y_2, \bar{\theta}),$$

where $z_2 = (x_2, y_2)$, $H_{\bar{\theta}} = \frac{1}{3}\sum_{i=1}^3 \nabla_\theta^2 \ell_0(x_i, y_i, \bar{\theta})$ is the Hessian, and $\ell_0(x_2, y_2, \bar{\theta}) = (\bar{\theta}^\top x_2 - y_2)^2$. IF then estimates the change $\Delta\theta$ ignoring the constraint by:

$$\theta(-\frac{1}{3}) - \theta(0) \approx -\frac{1}{3}\left.\frac{d\theta(\varepsilon)}{d\varepsilon}\right|_{\varepsilon=0}.$$

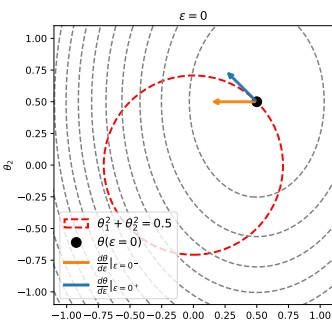 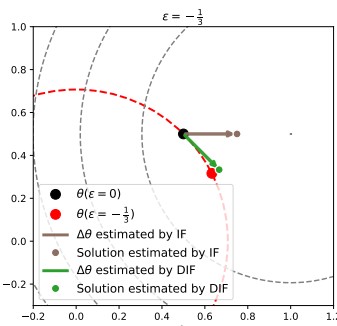

Figure 1: Loss landscape of the toy example when $\varepsilon = 0$ and $\varepsilon = -\frac{1}{3}$. The left plot illustrates the non-existence of the derivative of $\theta(\varepsilon)$ at $\varepsilon = 0$ as $\left.\frac{d\theta(\varepsilon)}{d\varepsilon}\right|_{\varepsilon=0^+} \neq \left.\frac{d\theta(\varepsilon)}{d\varepsilon}\right|_{\varepsilon=0^-}$. The right plot shows that the IF estimation of $\Delta\theta$ fails to account for the feasible region, whereas DIF accurately predicts the solution change direction and closely matches the ground truth.

However, as shown in the left panel of Figure 1, $\left.\frac{d\theta(\varepsilon)}{d\varepsilon}\right|_{\varepsilon=0}$ is not well defined because the left derivative and right derivative are inconsistent. When $\varepsilon > 0$, the point $\left(\frac{1}{2+3\varepsilon}, 0.5\right)$ satisfies $\|\cdot\|_2^2 < 0.5$ and thus lies in the interior of the feasible region. When $\varepsilon < 0$, $\|\theta(\varepsilon)\|_2^2$ exceeds 0.5, the solution is projected onto the $\ell_2$-boundary, resulting in two different one-sided derivatives at $\varepsilon = 0$. Moreover, the right panel of Figure 1 compares the ground truth $\Delta\theta$ with the change estimated by IF. When $\varepsilon$ shifts from 0 to $-\frac{1}{3}$, $\theta$ moves along the boundary of the feasible region, whereas the estimated change by the IF significantly deviates from the ground truth due to its failure to consider the constraint. In contrast, our estimator DIF (see Definition 2), which accounts for the geometry of the feasible region, closely matches the ground truth $\Delta\theta$ along the boundary. The derivation of DIF for this toy example is provided in Appendix D.

# 3 DIRECTIONAL INFLUENCE FUNCTION (DIF)

To address the non-smoothness of the solution map $\theta(\varepsilon)$ w.r.t. perturbation $\varepsilon$ and the inability of IF to handle constrained scenarios, this section proposes an alternative approach called the Directional Influence Function (DIF). We start with the definition of the classical IF.

**Definition 1** (Influence Function). The classical *IF* applies only to unconstrained problems. IF is defined as the derivative of $\theta$ w.r.t. the $\varepsilon_0$, i.e.,

$$IF(\bar{\theta}; Z^r) = \frac{d\theta(\varepsilon_0)}{d\varepsilon_0}\bigg|_{\varepsilon_0=0}. \tag{5}$$

**Definition 2** (Directional Influence Function). *DIF* of the data subset $Z^r$ is defined as the limiting directional derivative of the solution map $\theta(\varepsilon) : \mathbb{R}^{m+1} \to \mathbb{R}^d$ at $\bar{\varepsilon} = \mathbf{0}$, if the limit exists[2]

$$\text{DIF}\left(\bar{\theta}; Z^r\right) := D\theta(\bar{\varepsilon}; \Delta\bar{\varepsilon}) = \lim_{\substack{t\searrow 0^+ \\ \Delta\varepsilon\to\Delta\bar{\varepsilon}}} \frac{\theta(\bar{\varepsilon} + t\Delta\bar{\varepsilon}) - \theta(\bar{\varepsilon})}{t}, \tag{6}$$

where $\bar{\theta} = \theta(\bar{\varepsilon})$ denotes the model solution before perturbation. $\Delta\bar{\varepsilon} = \hat{\varepsilon} - \bar{\varepsilon} = \left[-\frac{1}{N_0}, -\frac{1}{N_1}, \dots, -\frac{1}{N_m}\right]$ is the perturbation direction, where $\Delta\bar{\varepsilon}$ corresponds to removing the data subset $Z^r$ from the constrained learning (1).

A function $f : X \to Y$ is called positively homogeneous if $f(\alpha x) = \alpha f(x) \quad \forall \alpha \geq 0, \forall x \in X$.

**Corollary 3.** *DIF is positively homogeneous,i.e.,*

$$\alpha D\theta(\bar{\varepsilon}; \Delta\bar{\varepsilon}) = D\theta(\bar{\varepsilon}; \alpha\Delta\bar{\varepsilon}), \quad \alpha \in \mathbb{R}^+$$

*Proof.* See Appendix C.4 for the full proof. $\square$

When $\varepsilon$ is perturbed from $\bar{\varepsilon}$ by a finite $\Delta\bar{\varepsilon}$, IF estimates $\Delta\theta$ via a first-order Taylor expansion. IF pertains to unconstrained learning and ignores constraints; thus

$$\Delta\theta \approx \frac{d\theta(\varepsilon_0)}{d\varepsilon_0}\bigg|_{\varepsilon=\bar{\varepsilon}} \cdot \Delta\bar{\varepsilon}_0. \tag{7}$$

This approximation, as demonstrated in the previous section, fails in the presence of constraints. Instead, DIF utilizes a directional derivative approach:

$$\Delta\theta \approx D\theta\left(\bar{\varepsilon}; \frac{\Delta\bar{\varepsilon}}{\|\Delta\bar{\varepsilon}\|}\right) \cdot \|\Delta\bar{\varepsilon}\| = D\theta\left(\bar{\varepsilon}; \Delta\bar{\varepsilon}\right). \tag{8}$$

where $D\theta(\bar{\varepsilon}; \frac{\Delta\bar{\varepsilon}}{\|\Delta\bar{\varepsilon}\|})$ is the directional derivative, and the second equation follows from Corollary 3. In what follows, we will demonstrate how to compute $D\theta\left(\bar{\varepsilon}; \Delta\bar{\varepsilon}\right)$ by solving a linearized VI system. We use $\Delta\hat{\theta}$ to denote the DIF-based estimate of $\Delta\theta$.

# 4 DERIVING DIF VIA SENSITIVITY ANALYSIS OF OPTIMALITY CONDITIONS

In this section, we show how DIF naturally arises from the sensitivity analysis of the optimality conditions of the constrained learning problem. Specifically, we first formulate the optimality system as a VI, then linearize this VI to obtain an auxiliary VI, and finally convert it into a QP whose solution yields the desired directional change in the model solution.

Constrained learning is typically solved using Lagrangian dual approaches, such as the augmented Lagrangian method, primal-dual methods (Chamon & Ribeiro, 2020; Ahmed et al., 2022; Nandwani et al., 2019). To quantify the impact of data downweighting on the solution of PCL problem (2), we

---

[2]This generalized definition accounts for potential non-smoothness of $\theta(\varepsilon)$ by allowing the direction to vary in a neighborhood of $\Delta\bar{\varepsilon}$, as is common in nonsmooth analysis(Clarke, 1990; Rockafellar & Wets, 1998).

introduce the perturbation $\varepsilon$ into the Lagrangian function, enabling us to analyze how the perturbation affects the optimality system. The Lagrangian function of PCL problem (2) is:

$$L(\varepsilon, \theta, \lambda) = \frac{1}{N_0} \sum_{i=1}^{N_0} \ell_0\left(z_i^{(0)}, \theta\right) + \varepsilon_0 \sum_{z_i^{(0)} \in Z^r} \ell_0\left(z_i^{(0)}, \theta\right)$$

$$+ \sum_{j=1}^{m} \lambda_j \left[ \frac{1}{N_j} \sum_{i=1}^{N_j} \ell_j\left(z_i^{(j)}, \theta\right) + \varepsilon_j \sum_{z_i^{(j)} \in Z^r} \ell_j\left(z_i^{(j)}, \theta\right) - \tau_j \right], \tag{9}$$

where $\lambda = [\lambda_1, \lambda_2, \ldots, \lambda_m], \lambda_j \geq 0$ denotes the dual variables. Let $(\theta(\varepsilon), \lambda(\varepsilon))$ be the primal–dual solution for a given $\varepsilon$. From Section 2, we use $(\bar{\theta}, \bar{\lambda}) = (\theta(\bar{\varepsilon}), \lambda(\bar{\varepsilon}))$ to denote the optimal solution of the constrained learning problem (1).

At $\theta = \bar{\theta}$, we partition the constraints into the active set $I_{\text{Active}}$ and the inactive set $I_{\text{Inactive}}$, where $I_{\text{Active}} = I_{\text{Binding}} \cup I_{\text{Non-binding}}$.

$$I_{\text{Active}} := \left\{ j \,\middle|\, \frac{1}{N_j} \sum_{i=1}^{N_j} \ell_j(z_i^{(j)}, \bar{\theta}) = \tau_j \right\}, \qquad I_{\text{Inactive}} := \left\{ j \,\middle|\, \frac{1}{N_j} \sum_{i=1}^{N_j} \ell_j(z_i^{(j)}, \bar{\theta}) < \tau_j \right\},$$

$$I_{\text{Binding}} := \{ j \in I_{\text{Active}} \mid \bar{\lambda}_j > 0 \}, \qquad I_{\text{Non-binding}} := \{ j \in I_{\text{Active}} \mid \bar{\lambda}_j = 0 \}.$$

Varying $\varepsilon$ from $\bar{\varepsilon}$ to $\hat{\varepsilon}$ corresponds to removing $Z^r$. Next, we study the sensitivity of the optimality conditions with respect to $\varepsilon$. We begin by examining the optimality condition of PCL (2).

### 4.1 Optimality Condition of PCL Problem (2)

The optimality condition of PCL (2) can be described by the Karush-Kuhn-Tucker (KKT) conditions. However, directly analyzing the KKT condition of (2) turns out to be difficult for understanding how the solution changes with data perturbations. Here, we adopt a VI formulation to characterize the optimality conditions. VI is a more general form of nonlinear optimization, which allows us to analyze solution perturbations through the lens of generalized differential calculus (Rockafellar & Wets, 1998). The optimality condition of PCL (2) is:

$$-\nabla_\theta L(\varepsilon, \theta, \lambda) \in N_{\mathbb{R}^d}(\theta), \quad \nabla_\lambda L(\varepsilon, \theta, \lambda) \in N_{\mathbb{R}^m_+}(\lambda), \tag{10}$$

where $N_{\mathbb{R}^d}(\theta), N_{\mathbb{R}^m_+}(\lambda)$ denote the Normal Cone.

**Definition 4** (Normal Cone; Dontchev & Rockafellar (2009)). Let $C \subseteq \mathbb{R}^d$ be a closed convex set. The normal cone to $C$ at a point $x \in C$ is defined as:

$$N_C(x) = \{v \in \mathbb{R}^d : \langle v, y - x \rangle \leq 0, \ \forall y \in C\}.$$

### 4.2 Linear Approximation of Variational Inequality

The classical derivation of IF relies on the Taylor expansion of the first-order optimality condition, which does not naturally handle constraints. In constrained learning, a natural choice for the optimality condition is the KKT system. Yet, estimating how $\varepsilon$ affects the KKT system is challenging due to its mixed system of equations and inequalities.

DIF adopts an alternative: representing the optimality condition as a VI (see formulation (10)) and applying a first-order approximation directly to the VI. The VI (10) can be compactly written as:

$$f(\varepsilon, \theta, \lambda) + N_E(\theta, \lambda) \ni \mathbf{0}, \tag{11}$$

$$\text{where} \quad f(\varepsilon, \theta, \lambda) = \begin{pmatrix} \nabla_\theta L(\varepsilon, \theta, \lambda) \\ -\nabla_\lambda L(\varepsilon, \theta, \lambda) \end{pmatrix}, E = \mathbb{R}^d \times \mathbb{R}^m_+$$

By definition, the solution $(\bar{\varepsilon}, \bar{\theta}, \bar{\lambda})$ satisfies the equation (11). When $\varepsilon$ is perturbed by $\Delta\bar{\varepsilon}$, estimating $\Delta\theta$ and $\Delta\lambda$ amounts to finding $\Delta\theta$ and $\Delta\lambda$ such that

$$f(\bar{\varepsilon} + \Delta\bar{\varepsilon}, \bar{\theta} + \Delta\theta, \bar{\lambda} + \Delta\lambda) + N_E(\bar{\theta} + \Delta\theta, \bar{\lambda} + \Delta\lambda) \ni \mathbf{0} \tag{12}$$

The linear approximation of (12) is

$$f(\bar{\varepsilon}, \bar{\theta}, \bar{\lambda}) + \nabla_\varepsilon f(\bar{\varepsilon}, \bar{\theta}, \bar{\lambda})\,\Delta\bar{\varepsilon} + \nabla_{(\theta,\lambda)} f(\bar{\varepsilon}, \bar{\theta}, \bar{\lambda}) \left[\begin{array}{c} \Delta\hat{\theta} \\ \Delta\hat{\lambda} \end{array}\right]$$

$$+ N_E((\bar{\theta}, \bar{\lambda}) + (\Delta\hat{\theta}, \Delta\hat{\lambda})) \ni \mathbf{0}. \tag{13}$$

Here, $(\Delta\hat{\theta}, \Delta\hat{\lambda})$, which satisfies (13), serves as an approximation of $(\Delta\theta, \Delta\lambda)$. Denote

$$A := \nabla_{(\theta,\lambda)} f(\bar{\varepsilon}, \bar{\theta}, \bar{\lambda}) = \left[\begin{array}{c} \nabla^2_\theta L(\bar{\varepsilon}, \bar{\theta}, \bar{\lambda}), \nabla_{\theta\lambda} L(\bar{\varepsilon}, \bar{\theta}, \bar{\lambda}) \\ -\nabla_{\theta\lambda} L(\bar{\varepsilon}, \bar{\theta}, \bar{\lambda}), -\nabla^2_\lambda L(\bar{\varepsilon}, \bar{\theta}, \bar{\lambda}) \end{array}\right],$$

$$\mu := \nabla_\varepsilon f(\bar{\varepsilon}, \bar{\theta}, \bar{\lambda}) = \left[\begin{array}{c} \nabla_{\theta\varepsilon} L(\bar{\varepsilon}, \bar{\theta}, \bar{\lambda}) \\ -\nabla_{\lambda\varepsilon} L(\bar{\varepsilon}, \bar{\theta}, \bar{\lambda}) \end{array}\right],$$

$$\Delta\hat{\eta} = \left[\begin{array}{c} \Delta\hat{\theta} \\ \Delta\hat{\lambda} \end{array}\right].$$

The linearized VI (13) can be expressed in the form:

$$f(\bar{\varepsilon}, \bar{\theta}, \bar{\lambda}) + \mu\Delta\bar{\varepsilon} + A\Delta\hat{\eta} + N_E((\bar{\theta}, \bar{\lambda}) + \Delta\hat{\eta}) \ni 0. \tag{14}$$

Here $\Delta\bar{\varepsilon}$ is given. Note that all terms in the linearized VI (14), except for $\Delta\hat{\eta}$—which is the variable we aim to estimate—can be precomputed from the known solution $\bar{\theta}$ and $\bar{\lambda}$. The linearized VI (14) can be simplified into the following Auxiliary VI (15).

**Proposition 5** (Auxiliary VI). *The solution $\Delta\hat{\eta}$ to the linearized VI (14) satisfies*

$$\mu\Delta\bar{\varepsilon} + A\Delta\hat{\eta} + N_K(\Delta\hat{\eta}) \ni 0, \tag{15}$$

*where $K = \mathbb{R}^d \times D$. $D$ is a space defined as*

$$D := \left\{ \Delta\hat{\lambda} \in \mathbb{R}^m \;\middle|\; \begin{array}{ll} \Delta\hat{\lambda}_j \in \mathbb{R} & \text{for } j \in I_{Binding}, \\ \Delta\hat{\lambda}_j \geq 0 & \text{for } j \in I_{Non\text{-}binding}, \\ \Delta\hat{\lambda}_j = 0 & \text{for } j \in I_{Inactive.} \end{array} \right\}.$$

*Proof.* See Appendix C.3 $\qquad\qquad\square$

Define (15) as the *Auxiliary VI*, which admits the same solution as the linearized VI (14), but with a simpler form.

**Proposition 6.** *Denote $\Delta\hat{\eta} = [\Delta\hat{\theta}; \Delta\hat{\lambda}]$ the solution to the Auxiliary VI (15). Then, $\Delta\hat{\theta}$ exactly recovers the directional derivative of the solution mapping $\theta(\varepsilon)$ at $\bar{\varepsilon}$ along $\Delta\bar{\varepsilon}$, i.e.,*

$$\Delta\hat{\theta} = D\theta(\bar{\varepsilon}; \Delta\bar{\varepsilon}). \tag{16}$$

*Proof.* See Appendix C.5. $\qquad\qquad\square$

**Remark 7.** By Proposition 6, computing DIF reduces to solving the Auxiliary VI (15).

The following theorem establishes the existence of DIF.

**Theorem 8.** *Assume the following regularity conditions hold:*

> **Assumption 1.** *(LICQ: Linear Independence Constraint Qualification) The gradients $\frac{1}{N_j}\sum_{i=1}^{N_j} \nabla_\theta \ell_j(z_i^{(j)}, \bar{\theta})$ for the active constraints ($j \in I_{Activate}$) are linearly independent.*

> **Assumption 2.** *(SOSC: Second-Order Sufficient Condition) For any $\Delta\theta \neq 0$ such that $\Delta\theta \perp \frac{1}{N_j}\sum_{i=1}^{N_j} \nabla_\theta \ell_j(z_i^{(j)}, \bar{\theta})$ for all $j \in I_{Binding}$, it holds that $\langle \Delta\theta, \nabla^2_{\theta\theta} L(\bar{\varepsilon}, \bar{\theta}, \bar{\lambda})\Delta\theta \rangle > 0$.*

*Then the directional derivative $D\theta(\bar{\varepsilon}; \Delta\bar{\varepsilon})$ exists.*

*Proof.* See Appendix C.2. $\qquad\qquad\square$

**Proposition 9.** *$\Delta\theta = \theta(\bar{\varepsilon} + \Delta\bar{\varepsilon}) - \theta(\bar{\varepsilon})$ is the ground truth and $\Delta\hat{\theta}$ is the estimation. Under the assumption of Theorem 8 , there exists a constant $M > 0$ such that, for all perturbations $\Delta\bar{\varepsilon}$ sufficiently small, the following holds:*

$$\|\Delta\theta - \Delta\hat{\theta}\| \le M\|\Delta\bar{\varepsilon}\| \tag{17}$$

*Proof.* See Appendix C.6. $\qquad\square$

Note that LICQ in Theorem 8 is the regular constraint qualification, while SOSC essentially requires some type of "local convexity" (i.e., at the current solution) of the Lagrangian function over all feasible directions that do not violate the binding constraints. They are the conditions that the proposed DIF method can apply to analyze data attributions of a learning model. This local convexity may exist even when the model is non-convex in general, such as deep learning models. Furthermore, Proposition 9 states that, under the two assumptions, solution changes over data perturbations are Lipschitz continuous. This ensures that when constructing the auxiliary problem (18) in the next subsection, the constraint set is stable, i.e., no sudden change will occur when infinitesimally small data perturbations are imposed.

### 4.3 Computing DIF via Quadratic Programming

Proposition 6 implies that computing the DIF reduces to solving the Auxiliary VI (15). In this section, we construct a QP (18) whose optimality is exactly the Auxiliary VI (15). By solving this QP, we can compute DIF and estimate the directional change in solution, i.e., $\Delta\theta$. Figure 2 summarizes the derivation in this section, illustrating the connections between the PCL problem (2), VI (11), Auxiliary VI (15), and QP (18) formulations.

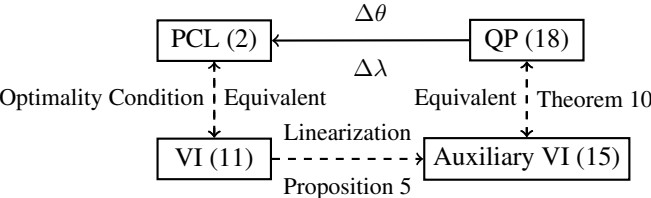

Figure 2: The relationship between (2) and (18).

We define the following QP:

$$\min_{\omega} \quad L(\bar{\varepsilon}, \bar{\theta}, \bar{\lambda}) + \left\langle \nabla_{\theta\varepsilon} L(\bar{\varepsilon}, \bar{\theta}, \bar{\lambda})\Delta\bar{\varepsilon}, \ \omega \right\rangle + \frac{1}{2}\left\langle \omega, \ \nabla^2_{\theta\theta} L(\bar{\varepsilon}, \bar{\theta}, \bar{\lambda})\,\omega \right\rangle$$

$$\text{s.t.} \quad \frac{1}{N_j}\sum_{i=1}^{N_j} \ell_j(z_i^{(j)}, \bar{\theta}) + \frac{1}{N_j}\sum_{i=1}^{N_j} \nabla_\theta \ell_j(z_i^{(j)}, \bar{\theta})\cdot\omega + \sum_{z_i^{(j)}\in Z^r} \Delta\bar{\varepsilon}_j \ell_j(z_i^{(j)}, \bar{\theta}) - \tau_j \begin{cases} = 0, & j \in I_{\text{binding}}, \\ \le 0, & j \in I_{\text{non-binding}}, \\ \text{free}, & j \in I_{\text{Inactive}}. \end{cases}$$
$$\tag{18}$$

**Theorem 10** (Auxiliary problem). *Let $(\omega^\star, \zeta^\star)$ denote the optimal primal-dual solution to the QP (18). Then $(\omega^\star, \zeta^\star)$ is also a solution to the auxiliary VI (15). In particular, the QP (18) and the auxiliary VI (15) admit the same solution pair under substitution $(\omega^*, \zeta^*) = (\Delta\hat{\theta}, \Delta\hat{\lambda})$.*

*Proof.* See Appendix C.7. $\qquad\square$

***Remark*** *11. By Proposition 6, $\Delta\hat{\theta} = D\theta(\bar{\varepsilon}; \Delta\bar{\varepsilon})$. Hence $\omega^\star$ is identical to $\text{DIF}(\bar{\theta}; Z^r)$.*

Finally, we establish the relationship between the IF and the DIF.

**Proposition 12.** *If no constraints are active at the solution $\bar{\theta}$, i.e., $I_{Active} = \emptyset$, then the DIF coincides with the classical IF.*

*Proof.* See Appendix C.8. $\qquad\square$

## 5 VALIDATION

### 5.1 DIF VALIDATING VIA CONSTRAINED LINEAR REGRESSION

Constrained linear regression is a widely applied learning model with hard constraints. It arises in several domains: in portfolio optimization, asset weights must be non-negative and sum to one, i.e., $\theta_i \geq 0, \sum_i \theta_i = 1$; in traffic flow modeling, flow variables must satisfy conservation laws and remain below capacity, e.g., $A\theta = 0, \theta \leq c$; and in fair machine learning, additional linear conditions are imposed to enforce equity across groups, such as $A_{\text{fair}} \theta \leq b_{\text{fair}}$.

This section leverages a constrained linear regression to evaluate the DIF estimator. We repeatedly remove one data point (100 trials) and compare the solution change $\Delta\theta$ predicted by DIF with the ground-truth retraining solution. We include the classical IF estimator and the penalty-based IF approximation as baselines.

**Data generation.** We generate a synthetic regression dataset. Specifically, we draw $n = 1000$ samples with $d = 5$ features:

$$X \in \mathbb{R}^{n \times d}, X_{ij} \sim \mathcal{N}(0,1), \quad \theta^* \sim \mathcal{N}(0, I_d), \quad y = X\theta^* + \varepsilon, \quad \varepsilon \sim \mathcal{N}\left(0, 0.1^2 I_n\right).$$

**Constrained Linear Regression.** The regression parameters are obtained by solving the constrained least-squares problem:

$$\hat{\theta} = \arg\min_{\theta \in \mathbb{R}^5} \frac{1}{2n} \|X\theta - y\|_2^2 \quad \text{s.t.} \quad A_{\text{eq}} \theta = b_{\text{eq}}, \quad A_{\text{ineq}} \theta \leq b_{\text{ineq}}$$

where

$$A_{\text{eq}} = [1,1,1,1,1], \quad b_{\text{eq}} = -4.0, \quad A_{\text{ineq}} = \begin{bmatrix} 1 & 1 & 0 & 0 & 0 \\ 0 & 0 & -1 & 0 & 1 \end{bmatrix}, \quad b_{\text{ineq}} = \begin{bmatrix} 1.5 \\ 1.5 \end{bmatrix}.$$

**Experimental results.** We perform 100 single-point removals. For each trial, we estimate the resulting change $\Delta\theta$ and compare it with the ground-truth leave-one-out (LOO) retraining. We leverage three estimators: i) The IF estimates the solution change by ignoring the constraints, yielding $\Delta\theta_{\text{IF}}$ as in equation (7). ii) The penalty-based IF adds soft penalties for the constraints to the objective and applies the IF estimator on the penalized surrogate. iii) The DIF enforces feasibility by solving the QP (18), yielding $\Delta\theta_{\text{DIF}}$ as in equation (8). Appendix E provides the full derivations for all three methods in the constrained linear regression setting. All experiments are solved with `CVXPY`. As shown in Figure 3, DIF nearly coincides with LOO (points align with $y=x$), whereas IF and penalty IF exhibit noticeable biases.

### 5.2 DIF VALIDATION VIA CONSTRAINED CNN

**Model** We adopt the following constrained learning formulation (Shen et al., 2022):

$$\min_{\theta \in \Theta} \overline{R}(f_\theta) \quad \text{s.t.} \quad R_i(f_\theta) - \overline{R}(f_\theta) - \tau \leq 0, \ i = 1, \dots, m, \tag{19}$$

where $R_i(f_\theta) = \mathbb{E}_{(x,y) \sim \mathcal{D}_i}[\ell(f_\theta(x), y)]$ is the risk of the client (or group) $i$, and $\overline{R}(f_\theta) = \frac{1}{C} \sum_{i=1}^C R_i(f_\theta)$ is the average risk. The constraints ensure that no group's risk exceeds the global average risk by more than $\tau$, thereby limiting the performance disparity across groups.

We use a network with seven convolutional layers and $\tanh(\cdot)$ non-linearities, modeled after the all convolutional network of Springenberg et al. (2014). We train it on the MNIST training set (LeCun et al., 1998). We solve this problem using a primal–dual method, jointly updating the model parameters $\theta$ and the Lagrange multipliers.

**Heterogeneous Data Partitioning.** We create non-IID group-wise data partitions following Shen et al. (2022). We split the dataset into $C$ groups by allocating an $\alpha$-fraction of the data uniformly at random and distributing the remaining $(1 - \alpha)$-fraction in a label-skewed way, where samples are sorted by class labels and assigned consecutively to groups. Unless otherwise specified, we set the number of groups to $m = 3$ and $\alpha = 0.5$. This setup creates label-imbalanced distributions across different groups.

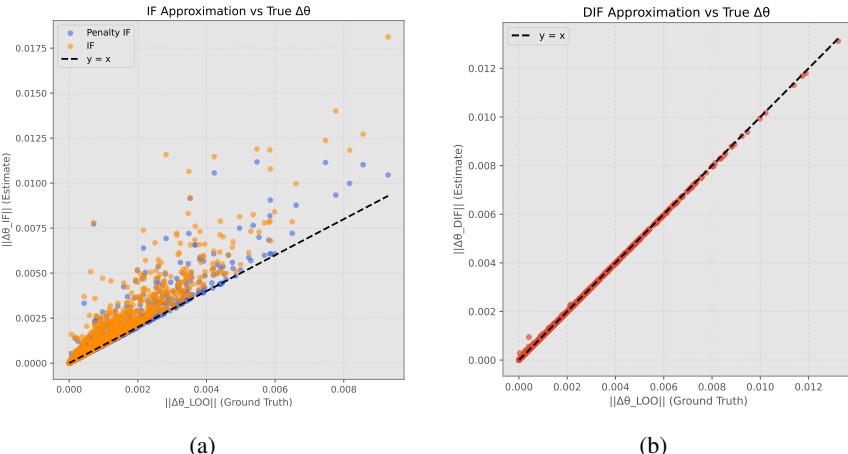

(a)          (b)

Figure 3: Comparison of solution changes on a constrained linear regression task. (a) IF and the penalty-based IF estimators versus the ground-truth leave-one-out retraining results. (b) Estimation of the proposed DIF versus the same ground-truth values. DIF aligns almost perfectly with the $y = x$ line, demonstrating its ability to accurately capture parameter changes under constraints.

**Influence Estimation.** We employ DIF to estimate the effect of removing training samples under the fairness constraint. We select the 100 most influential training samples and remove one sample at a time (100 trials in total). Since CNNs are typically non-convex, the solution may shift to another valley in the loss landscape during retraining. Therefore, instead of directly comparing the predicted solution change $\Delta\theta_{\mathrm{DIF}}$ with the ground-truth change $\Delta\theta_{\mathrm{LOO}}$, we evaluate DIF indirectly by comparing their resulting loss changes on a misclassified test point. Specifically, we approximate the updated model as $\theta' \approx \theta + \Delta\theta_{\mathrm{DIF}}$ and compute the predicted loss difference $\Delta\ell = R\big(f_{\theta'}(x_{\mathrm{test}}), y_{\mathrm{test}}\big) - R\big(f_\theta(x_{\mathrm{test}}), y_{\mathrm{test}}\big)$. Figure 4 compares these DIF-predicted loss differences with the actual loss differences by retraining the model. The points align closely with the $y = x$ line (Pearson $r = 0.90$), indicating that DIF accurately predicts the influence of individual training samples on this test loss. This indicates that although CNN is non-convex in general, the local convexity as in Theorem 8 might still hold such that DIF still performs well.

We further compute the penalty-IF predicted loss differences on the same 100 influential training samples used in our DIF evaluation. We observe that: 1) the standard IF without any penalty does not converge in practice. The conjugate gradient (CG) method used to compute the Hessian–vector products (HVPs) hardly converges even after carefully tuning the damping parameters, because the curvature becomes ill-conditioned under the fairness constraints. 2) for the penalty-IF, the CG method is able to converge after incorporating constraints into the objective through a penalized surrogate. However, the predicted loss differences become very sensitive to the choice of the penalty parameter. Changing the penalty value by only $0.01$ can noticeably change the predicted loss difference. 3) for penalty-IF, after tuning the penalty value (we set it to $k = 0.06$), the

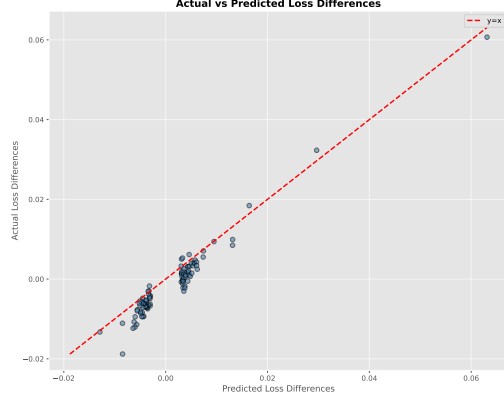

Figure 4: Actual vs. DIF predicted loss differences on a misclassified test sample.

penalty-IF reaches a Pearson correlation of $r \approx 0.62$ between the predicted and actual LOO loss differences. This result is still significantly lower than that of DIF $r = 0.90$. This indicates that the penalty formulation does not match the true KKT-based sensitivity structure. Table 1 summarizes the numerical results for 10 randomly selected cases from the full set of 100 samples.

Table 1: Comparison of IF predicted influence, DIF result, and actual retraining loss change.

| Sample ID | Predicted (Penalty-IF) | Predicted (DIF) | Actual (Retrain) |
|---|---|---|---|
| 4221 | 0.005528 | 0.004979 | 0.003664 |
| 215 | 0.000016 | 0.004249 | 0.003015 |
| 1328 | -0.009318 | -0.003962 | -0.003139 |
| 4668 | 0.013700 | 0.003922 | 0.004601 |
| 772 | 0.001171 | 0.003030 | 0.003509 |
| 4473 | 0.005000 | 0.002985 | 0.002487 |
| 4144 | -0.007275 | -0.002820 | -0.000704 |
| 2828 | 0.002719 | 0.002775 | 0.002441 |
| 3505 | 0.001118 | 0.002747 | 0.003592 |
| 1163 | 0.001183 | 0.002359 | 0.002551 |

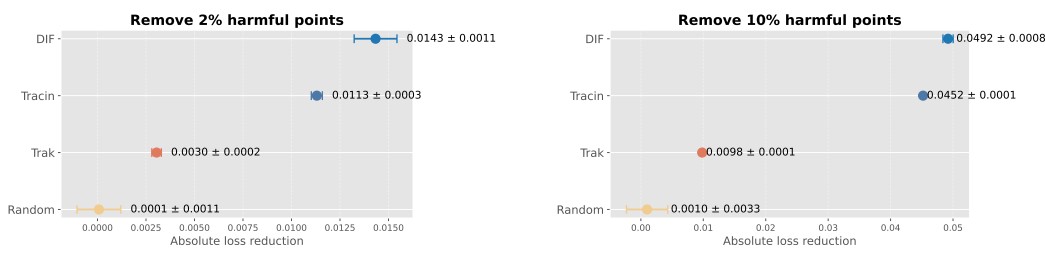

(a) Remove 2% harmful points      (b) Remove 10% harmful points

Figure 5: Absolute loss reduction obtained by deleting the most harmful samples selected by different methods. Dot represents the mean loss reduction and the horizontal bar is the standard deviation.

**Counterfactual retraining.** We evaluate counterfactual retraining using three influence estimators-DIF, TracIn (Pruthi et al., 2020), and TRAK (Park et al., 2023). For each method, we compute per-sample influence scores on the training set, remove the most harmful $2\%$ and $10\%$ samples, and then retrain the model. We compare the resulting loss changes on a fixed misclassified test point. The constrained CNN is trained using a primal-dual method. For TracIn, we develop a constrained variant following prior work. During primal-dual updates, we record the gradients of the Lagrangian loss with respect to the model parameters. The TracIn score for a training-test pair is computed as $\text{TracInCP}\,(x_{\text{train}}\,, x_{\text{test}}\,) = \sum_{i=1}^{k} \delta_i \nabla \ell_{\text{lag}}\,(\theta_{t_i}, x_{\text{train}}\,) \cdot \nabla \ell_{\text{lag}}\,(\theta_{t_i}, x_{\text{test}}\,)$, where $\delta_i$ denotes the step size between checkpoints $i-1$ and $i$, and $\{\theta_{t_1}, \theta_{t_2}, \ldots, \theta_{t_k}\}$ are the parameter checkpoints recorded along the primal-dual trajectory. For TRAK, we apply the method to the penalized objective, where the constraints are incorporated into the loss via a squared hinge penalty.

As shown in Figure 5, comparing with removing points randomly, DIF and TracIn achieve the highest loss reductions with DIF slightly outperforming TracIn. This indicates that both methods identify the most harmful samples in our constrained training setup. In contrast, TRAK provides little improvement. This is because its random projection–based approximation fails to preserve the constraint-adjusted gradient directions, resulting in nearly ineffective influence estimates.

## 6 CONCLUSION

This paper introduces DIF as a data attribution method for constrained learning. Methodologically, we develop a VI-based sensitivity analysis framework for constrained learning and use it to derive a first-order, feasibility-preserving estimator of solution changes. Empirically, on constrained linear regression and fairness-constrained CNNs, DIF closely tracks leave-one-out retraining, whereas IF and penalty-based IF exhibit substantial bias or infeasibility. Looking ahead, our DIF framework may be applied to data poisoning attacks, machine unlearning, and online or distribution-shifted training with evolving constraint sets.

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

APPENDIX

# A    RELATED WORK

IF was first introduced by Hampel (1974) to study the robustness of statistical estimators under distributional shifts. Decades later, Koh & Liang (2017) adapted IF to machine learning to analyze the impact of individual training samples on model behavior. Since then, numerous extensions have emerged, including faster and more scalable inference methods (Schioppa et al., 2022; Guo et al., 2020), and extensions to domains such as time-series forecasting (Zhang et al., 2024) and generative modeling (Terashita et al., 2021; Kong & Chaudhuri, 2021; Lin et al., 2024). Beyond these, IF has been applied to practical tasks such as defending against data poisoning (Steinhardt et al., 2017), improving interpretability (Han et al., 2020), and supporting machine unlearning (Wang et al., 2025; Warnecke et al., 2021). However, because IF fundamentally relies on the implicit function theorem (Krantz & Parks, 2002), most methods require strong convexity of the model. Recent empirical studies (Bae et al., 2022) further highlight this limitation, showing that IF estimates can be fragile and often misaligned with their actual effects in practice (Basu et al., 2020; Ghorbani et al., 2019).

Constrained learning has gained increasing attention due to the need for machine learning models that are fair (Quadrianto et al., 2019; Mehrabi et al., 2021), robust (Zhang et al., 2019), and consistent with domain-specific priors (Borghesi et al., 2020). For instance, invariant learning aims to ensure stable performance under distribution shifts or data transformations Benton et al. (2020); Immer et al. (2022), and PINNs incorporate traffic flow theory into transportation forecasting Shi et al. (2021). These settings highlight a growing class of learning problems where solutions must satisfy explicit feasibility requirements beyond minimizing prediction (Donini et al., 2018; Robey et al., 2021; Hounie et al., 2023; Li et al., 2023; Shen et al., 2022). Although adding a regularization or penalty term to the objective function is common practice in ML, this approach suffers from inherent limitations. First, the choice of trade-off parameters is highly non-trivial, often requiring heuristic tuning without guarantees of correctness. Second, the gradients of the prediction loss and the penalty term rarely align; as a result, gradient-based optimization must continually balance these two gradient directions, which may lead to suboptimal updates (Hwang & Lim, 2024; Liu et al., 2024).

Given the wide application of constrained learning, it is essential to develop data attribution methods that explicitly account for feasibility. Classical influence-function analyses have considered constraints, but only in limited settings. Ghosh (2014) studied restricted minimum-divergence estimators and derived closed-form IF expressions under smooth equality constraints, while Vasconcellos & Fernandez (2009) extended Cook's local-influence framework to perturbations constrained by homogeneous linear restrictions. In this paper, we extend the classical IF framework to general constrained learning by analyzing solution changes within the feasible region. Specifically, we reformulate the optimality conditions as a VI and study its sensitivity with respect to data weights. This formulation naturally admits both primal and dual sensitivities, allowing us to capture how perturbations affect not only the model parameters but also the associated Lagrangian multipliers. Compared to the KKT system, the VI method offers a more general analysis framework, together with a more compact geometric language for describing the coupled dynamics of primal and dual variables by characterizing solution changes within the tangent cone of the feasible region.

# B    PRELIMINARY

**Definition 13** (Polar Cone; cf. Dontchev & Rockafellar (2009)). Let $K \subseteq \mathbb{R}^n$ be a closed convex cone. The *polar cone* of $K$ is defined as

$$K^* = \{\, y \in \mathbb{R}^n \ \mid \ \langle x, y \rangle \leq 0 \ \ \forall x \in K \,\}. \tag{20}$$

Moreover, the normal vectors to $K$ and $K^*$ satisfy

$$y \in N_K(x) \iff x \in N_{K^*}(y) \iff x \in K, \ y \in K^*, \ \langle x, y \rangle = 0. \tag{21}$$

**Definition 14** (Tangent Cone, cf. Rockafellar & Wets (1998)). For a set $C \subset \mathbb{R}^n$ (not necessarily convex) and a point $x \in C$, a vector $v$ is said to be tangent to $C$ at $x$ if

$$\frac{1}{\tau^k}(x^k - x) \to v \quad \text{for some } x^k \to x, \ x^k \in C, \ \tau^k \downarrow 0. \tag{22}$$

The set of all such vectors $v$ is called the *tangent cone* to $C$ at $x$ and is denoted by $T_C(x)$. For $x \notin C$, we take $T_C(x) = \emptyset$.

**Definition 15** (Critical Cone, cf. Rockafellar & Wets (1998)). For a convex set $C$, any $x \in C$ and any $v \in N_C(x)$, the *critical cone* to $C$ at $x$ for $v$ is

$$K_C(x, v) = \{ w \in T_C(x) \mid w \perp v \}. \tag{23}$$

**Definition 16** (Critical Subspaces). For a convex set $C \subset \mathbb{R}^n$ and $(x, v)$ with $v \in N_C(x)$, let $K_C(x, v)$ denote the critical cone at $(x, v)$. Then the associated *critical subspaces* are defined as

$$K_C^+(x, v) \ = \ K_C(x, v) - K_C(x, v) \ = \ \{ w - w' \mid w, w' \in K_C(x, v) \}, \tag{24}$$

$$K_C^-(x, v) \ = \ K_C(x, v) \cap [-K_C(x, v)] \ = \ \{ w \in K_C(x, v) \mid -w \in K_C(x, v) \}. \tag{25}$$

Here $K_C^+(x, v)$ is the smallest linear subspace containing $K_C(x, v)$, and $K_C^-(x, v)$ is the largest linear subspace contained in $K_C(x, v)$.

**Lemma 17** (Reduction Lemma;cf. 2E.4 Dontchev & Rockafellar (2009)). *Let $C \subset \mathbb{R}^n$ be a convex set and $\bar{x} \in C$ with $\bar{v} \in N_C(\bar{x})$. Define the critical cone $K_C := K_C(\bar{x}, \bar{v})$. Then, for all sufficiently small $w, u \in \mathbb{R}^n$, we have the equivalence*

$$\bar{v} + \Delta v \in N_C(\bar{x} + \Delta x) \iff \Delta v \in N_{K_C}(\Delta x). \tag{26}$$

## C  PROOFS

### C.1  AUXILIARY LEMMAS

For clarity, we first introduce some notation and rewrite the generalized equation in a standard form.

**Notation.** Let $\eta = (\theta, \lambda)$ denote the parameter vector, $\bar{\eta} = (\bar{\theta}, \bar{\lambda})$ be its reference value, and $\Delta \hat{\eta} = (\Delta \hat{\theta}, \Delta \hat{\lambda})$ be the estimation to $\Delta \eta = (\Delta \theta, \Delta \lambda)$.

[**JB**: here add a definition of VI; and some general comments such as VI is a more general form of optimization problems]

**Solution mapping.** We define the solution mapping as

$$S(\varepsilon) := \{ (\theta, \lambda) \mid f(\varepsilon, \theta, \lambda) + N_E(\theta, \lambda) \ni 0 \} = \{ \eta \mid f(\varepsilon, \eta) + N_E(\eta) \ni 0 \}, \tag{27}$$

where $N_E(\eta)$ denotes the normal cone to $E$ at $\eta$.

**Generalized equation form.** To study the local behavior of $S$, we rewrite the system as the generalized equation

$$G(\eta) := f(\bar{\varepsilon}, \bar{\eta}) + \nabla_\eta f(\bar{\varepsilon}, \bar{\eta})(\eta - \bar{\eta}) + N_E(\eta), \tag{28}$$

with

$$\nabla_\eta f(\bar{\varepsilon}, \bar{\eta}) = \nabla_{(\theta, \lambda)} f(\bar{\theta}, \bar{\lambda}, \bar{x}) =: A. \tag{29}$$

At the reference point, the optimality condition (11) ensures that

$$G(\bar{\eta}) \ni 0. \tag{30}$$

**Linearization.** We then consider the linearized generalized equation (the first-order approximation of $G$):

$$G_0(\Delta \eta) := \nabla_\eta f(\bar{\varepsilon}, \bar{\eta}) \Delta \eta + N_K(\Delta \eta) = A \Delta \eta + N_K(\Delta \eta), \qquad G_0(0) \ni 0, \tag{31}$$

where $K := K_E(\bar{\eta}, -f(\bar{\varepsilon}, \bar{\eta}))$ is the critical cone. This coincides with $K = \mathbb{R}^d \times D$ as introduced in Proposition 5.

It is convenient to define the inverse-type mapping

$$\bar{s} := G_0^{-1} = (A + N_K)^{-1}, \tag{32}$$

which solves the linearized inclusion (31).

Given perturbation $u \Delta \bar{\varepsilon}$,

$$\bar{s}(-u \Delta \bar{\varepsilon}) = (A + N_K)^{-1}(-u \Delta \bar{\varepsilon}) = \{ \Delta \hat{\eta} \mid A \Delta \hat{\eta} + N_K(\Delta \hat{\eta}) + u \Delta \bar{\varepsilon} \ni 0 \}. \tag{33}$$

In other words, $\bar{s}(-u \Delta \bar{\varepsilon})$ yields the solution to the auxiliary VI (15).

**Critical subspace of $E$.** Following the definition 16, the critical subspaces of $E$ at $(\bar{\theta}, \bar{\lambda}, -f(\bar{\varepsilon}, \bar{\theta}, \bar{\lambda}))$ are defined as

$$K_E^+(\bar{\theta}, \bar{\lambda}, -f(\bar{\varepsilon}, \bar{\theta}, \bar{\lambda})) = \mathbb{R}^d \times K_{\mathbb{R}^m}^+(\bar{\lambda}, \nabla_\lambda L(\bar{\varepsilon}, \bar{\theta}, \bar{\lambda})), \tag{34}$$

$$K_E^-(\bar{\theta}, \bar{\lambda}, -f(\bar{\varepsilon}, \bar{\theta}, \bar{\lambda})) = \mathbb{R}^d \times K_{\mathbb{R}^m}^-(\bar{\lambda}, \nabla_\lambda L(\bar{\varepsilon}, \bar{\theta}, \bar{\lambda})). \tag{35}$$

$$\Delta\hat{\eta} = (\Delta\hat{\theta}, \Delta\hat{\lambda}) \text{ satisfies } \Delta\hat{\eta} \in K_E^+ \iff \Delta\hat{\lambda}_j = 0, \forall j \in I_{\text{Inactive}}, \text{ and } \Delta\hat{\theta} \in \mathbb{R}^d. \tag{36}$$

**Lemma 18.** *Under the assumptions of Theorem 8, for any $\Delta\hat{\eta}$ satisfying the linearized VI (15), if*

$$\Delta\hat{\eta} \in K_E^+, \quad \Delta\hat{\eta} \neq 0, \quad A\Delta\hat{\eta} \perp K_E^-, \tag{37}$$

*then*

$$\langle \Delta\hat{\eta}, A\Delta\hat{\eta} \rangle > 0.$$

*Proof.* For any $\Delta\hat{\eta}$ satisfying the linearized VI (15), it follows from the definition of the space $D$ that $\Delta\hat{\lambda}_j = 0$ for all $j$ in the inactive set. Therefore, any $\Delta\hat{\eta}$ satisfying VI (15) also satisfies $\Delta\hat{\eta} \in K_E^+$.

We next analyze the orthogonality condition $A\Delta\hat{\eta} \perp K_E^-$. By expanding $A(\Delta\hat{\theta}, \Delta\hat{\lambda})$ and splitting the $\theta$- and $\lambda$-parts, we obtain:

$$A\Delta\hat{\eta} \perp K_E^- \iff A(\Delta\hat{\theta}, \Delta\hat{\lambda}) \perp K_E^-$$

$$\iff \left( H\Delta\hat{\theta} + \nabla_{\theta\lambda}L(\bar{\varepsilon}, \bar{\theta}, \bar{\lambda})\Delta\hat{\lambda}, \ -\nabla_{\lambda\theta}L(\bar{\varepsilon}, \bar{\theta}, \bar{\lambda})\Delta\hat{\theta} \right) \perp K_E^-$$

$$\iff H\Delta\hat{\theta} + \nabla_{\theta\lambda}L(\bar{\varepsilon}, \bar{\theta}, \bar{\lambda})\Delta\hat{\lambda} \perp \mathbb{R}^d, \quad -\nabla_{\lambda\theta}L(\bar{\varepsilon}, \bar{\theta}, \bar{\lambda})\Delta\hat{\theta} \perp K_{\mathbb{R}^m}^-(\bar{\lambda}, \nabla_\lambda L(\bar{\varepsilon}, \bar{\theta}, \bar{\lambda}))$$

$$\iff H\Delta\hat{\theta} + \nabla_{\theta\lambda}L(\bar{\varepsilon}, \bar{\theta}, \bar{\lambda})\Delta\hat{\lambda} = 0, \quad -\nabla_{\lambda\theta}L(\bar{\varepsilon}, \bar{\theta}, \bar{\lambda})\Delta\hat{\theta} \perp K_{\mathbb{R}^m}^-(\bar{\lambda}, \nabla_\lambda L(\bar{\varepsilon}, \bar{\theta}, \bar{\lambda})).$$

Moreover, by the definition of the critical subspace, we have

$$\Delta\hat{\lambda} \in K_{\mathbb{R}^m}^-(\bar{\lambda}, \nabla_\lambda L(\bar{\varepsilon}, \bar{\theta}, \bar{\lambda})) \iff \Delta\hat{\lambda}_j = 0, \forall j \in I_{\text{non-binding}} \cup I_{\text{Inactive}}, \tag{38}$$

where $I_{\text{Inactive}}$ and $I_{\text{non-binding}}$ denote the sets of inactive and non-binding constraints, respectively.

Consequently, the condition

$$-\nabla_{\lambda\theta}L(\bar{\varepsilon}, \bar{\theta}, \bar{\lambda})\Delta\hat{\theta} \ \perp \ K_{\mathbb{R}^m}^-(\bar{\lambda}, \nabla_\lambda L)$$

only requires that

$$\nabla_{\lambda_j\theta}L\Delta\hat{\theta} = 0, \qquad \forall j \in I_{\text{binding}}.$$

That is,

$$A\Delta\hat{\eta} \perp K_E^- \iff H\Delta\hat{\theta} + \nabla_{\theta\lambda}L(\bar{\varepsilon}, \bar{\theta}, \bar{\lambda})\Delta\hat{\lambda} = 0, \quad \nabla_{\lambda_j\theta}L\Delta\hat{\theta} = 0, \forall j \in I_{\text{binding}}. \tag{39}$$

Notice that

$$\nabla_{\lambda_j\theta}L\Delta\hat{\theta} = 0 \iff \Delta\hat{\theta} \perp \frac{1}{N_j}\sum_{i=1}^{N_j}\nabla_\theta\ell_j\left(z_i^{(j)}, \bar{\theta}\right), \qquad \forall j \in I_{\text{binding}}. \tag{40}$$

Assumption 2 in Theorem 8 implies that for all $\Delta\hat{\theta}$ satisfying (39), we have

$$\left\langle \Delta\hat{\theta}, \nabla_{\theta\theta}^2 L(\bar{\varepsilon}, \bar{\theta}, \bar{\lambda})\Delta\hat{\theta} \right\rangle > 0. \tag{41}$$

Moreover, substituting

$$A := \nabla_{(\theta, \lambda)}f(\bar{\varepsilon}, \bar{\theta}, \bar{\lambda}) = \begin{bmatrix} \nabla_\theta^2 L(\bar{\varepsilon}, \bar{\theta}, \bar{\lambda}) & \nabla_{\theta\lambda}L(\bar{\varepsilon}, \bar{\theta}, \bar{\lambda}) \\ -\nabla_{\theta\lambda}L(\bar{\varepsilon}, \bar{\theta}, \bar{\lambda}) & -\nabla_\lambda^2 L(\bar{\varepsilon}, \bar{\theta}, \bar{\lambda}) \end{bmatrix},$$

to (41),

we obtain

$$\langle \Delta\hat{\eta}, \, A\Delta\hat{\eta} \rangle = \langle (\Delta\hat{\theta}, \Delta\hat{\lambda}), \, A(\Delta\hat{\theta}, \Delta\hat{\lambda}) \rangle = \langle \Delta\hat{\theta}, \, \nabla^2_{\theta\theta} L(\bar{\varepsilon}, \bar{\theta}, \bar{\lambda}) \, \Delta\hat{\theta} \rangle. \tag{42}$$

Combining the two results yields $\langle \Delta\hat{\eta}, A\Delta\hat{\eta} \rangle > 0$.

$\square$

**Lemma 19.** *Under the assumptions of Theorem 8, $\bar{s} := (A + N_K)^{-1}$ is everywhere single-valued. Equivalently, there exists a unique solution $\Delta\hat{\eta}$ to the auxiliary VI (15) given $(\bar{\varepsilon}, \bar{\theta}, \bar{\lambda})$ and $\Delta\hat{\varepsilon}$.*

*Proof.* We prove that $G_0^{-1} := (A + N_K)^{-1}$ is single-valued everywhere.

Assume, for contradiction, that there exist two distinct solutions $\Delta\hat{\eta}_1$ and $\Delta\hat{\eta}_2$ such that $r = G_0^{-1}(\Delta\hat{\eta}_1) = G_0^{-1}(\Delta\hat{\eta}_2)$. Then,

$$\Delta\hat{\eta}_1, \Delta\hat{\eta}_2 \in K, \qquad r - A\Delta\hat{\eta}_1 \in N_K(\Delta\hat{\eta}_1), \qquad r - A\Delta\hat{\eta}_2 \in N_K(\Delta\hat{\eta}_2).$$

This implies, by the definition of the normal cone, that

$$\langle \Delta\hat{\eta}_1, \, r - A\Delta\hat{\eta}_2 \rangle \le 0, \qquad \langle \Delta\hat{\eta}_2, \, r - A\Delta\hat{\eta}_1 \rangle \le 0. \tag{43}$$

Note that $K$ is the critical cone. By condition (21), we have

$$\Delta\hat{\eta}_1 \in K, \quad r - A\Delta\hat{\eta}_1 \in K^*, \quad \langle \Delta\hat{\eta}_1, \, r - A\Delta\hat{\eta}_1 \rangle = 0,$$
$$\Delta\hat{\eta}_2 \in K, \quad r - A\Delta\hat{\eta}_2 \in K^*, \quad \langle \Delta\hat{\eta}_2, \, r - A\Delta\hat{\eta}_2 \rangle = 0. \tag{44}$$

By the definition of the polar cone $K^* = (K^-)^\perp$, we also have

$$-A(\Delta\hat{\eta}_1 - \Delta\hat{\eta}_2) \in K^* - K^* = (K^-)^\perp, \qquad \Delta\hat{\eta}_1 - \Delta\hat{\eta}_2 \in K - K = K^+.$$

Consider the following inner product:

$$\begin{aligned}
\langle \Delta\hat{\eta}_1 - \Delta\hat{\eta}_2, \, A(\Delta\hat{\eta}_1 - \Delta\hat{\eta}_2) \rangle &= \langle \Delta\hat{\eta}_1 - \Delta\hat{\eta}_2, \, [r - A\Delta\hat{\eta}_2] - [r - A\Delta\hat{\eta}_1] \rangle \\
&= \langle \Delta\hat{\eta}_1, \, r - A\Delta\hat{\eta}_2 \rangle - \langle \Delta\hat{\eta}_1, \, r - A\Delta\hat{\eta}_1 \rangle \\
&\quad - \langle \Delta\hat{\eta}_2, \, r - A\Delta\hat{\eta}_2 \rangle + \langle \Delta\hat{\eta}_2, \, r - A\Delta\hat{\eta}_1 \rangle \\
&\le 0.
\end{aligned}$$

Following Lemma 18, Assumption 2 of Theorem 8 ensures that for any nonzero $\Delta\hat{\eta} \in K^+$ with $A\Delta\hat{\eta} \perp K^-$, we must have $\langle \Delta\hat{\eta}, \, A\Delta\hat{\eta} \rangle > 0$. This contradicts the above inequality unless $\Delta\hat{\eta}_1 = \Delta\hat{\eta}_2$.

Therefore, the solution must be unique, which proves that $G_0^{-1} = (A + N_K)^{-1}$ is everywhere single-valued. $\square$

***Remark*** 20. According to Dontchev and Rockafellar (Dontchev & Rockafellar, 2009, Chapter 2E), consider the generalized equation $G_0 = A + N_K$, where $A$ is a linear operator and $N_K$ is the normal cone mapping of a closed convex cone $K$. If the inverse mapping $G_0^{-1}$ is *everywhere single-valued*, then $G_0^{-1}$ is (globally) Lipschitz continuous.

**Lemma 21** (Relation between $G^{-1}$ and $G_0^{-1}$). *Then for all $\Delta\eta$ sufficiently close to $0$, the inverse mappings $G^{-1}$ and $G_0^{-1}$ satisfy the following relationship:*

$$G^{-1}(\gamma) = G_0^{-1}(\gamma) + \bar{\eta}, \qquad \gamma \text{ near } 0. \tag{45}$$

*Proof.* Consider any $\Delta\eta$ close to $0$. We first expand the generalized equation $G(\bar{\eta} + \Delta\eta)$ as

$$\begin{aligned}
G(\bar{\eta} + \Delta\eta) &= f(\bar{\varepsilon}, \bar{\eta}) + \nabla_\eta f(\bar{\varepsilon}, \bar{\eta}) \, \Delta\eta + N_E(\bar{\eta} + \Delta\eta) \\
&= f(\bar{\varepsilon}, \bar{\eta}) - \nabla_\eta f(\bar{\varepsilon}, \bar{\eta}) \, \bar{\eta} + \nabla_\eta f(\bar{\varepsilon}, \bar{\eta})(\bar{\eta} + \Delta\eta) + N_E(\bar{\eta} + \Delta\eta).
\end{aligned}$$

Let $A := \nabla_\eta f(\bar\varepsilon, \bar\eta)$. Then $\gamma \in G(\bar\eta + \Delta\eta)$ if and only if

$$\gamma \in f(\bar\varepsilon, \bar\eta) - A\bar\eta + A(\bar\eta + \Delta\eta) + N_E(\bar\eta + \Delta\eta). \tag{46}$$

Rearranging the terms, this is equivalent to

$$\gamma - f(\bar\varepsilon, \bar\eta) + A\bar\eta \in A(\bar\eta + \Delta\eta) + N_E(\bar\eta + \Delta\eta), \tag{47}$$

which can be further rewritten as

$$\gamma - f(\bar\varepsilon, \bar\eta) \in A\Delta\eta + N_E(\bar\eta + \Delta\eta). \tag{48}$$

Note that $-f(\bar\varepsilon, \bar\eta) \in N_E(\bar\eta)$. By the *Reduction Lemma* 17, this holds if and only if

$$\gamma \in A\Delta\eta + N_K(\Delta\eta), \tag{49}$$

which is exactly

$$\gamma \in G_0(\Delta\eta). \tag{50}$$

$\square$

**Lemma 22.** *Define*

$$\sigma(\varepsilon) := G^{-1}\big(-\nabla_\varepsilon f(\bar\varepsilon, \bar\eta)\,(\varepsilon - \bar\varepsilon)\big) = G^{-1}\big(-u(\varepsilon - \bar\varepsilon)\big).$$

*If $G^{-1}$ admits a single-valued Lipschitz localization around $0$ for $\bar\eta$, then $\sigma(\varepsilon)$ is a first-order approximation of $S(\varepsilon)$ at $\bar\varepsilon$.*

*Proof.* Since $f$ is strictly differentiable at $(\bar\varepsilon, \bar\eta)$, for any small perturbations $(\Delta\varepsilon, \Delta\eta)$ we have

$$f(\bar\varepsilon + \Delta\varepsilon, \bar\eta + \Delta\eta) = f(\bar\varepsilon, \bar\eta) + \nabla_\varepsilon f(\bar\varepsilon, \bar\eta)\,\Delta\varepsilon + \nabla_\eta f(\bar\varepsilon, \bar\eta)\,\Delta\eta + o(\Delta\varepsilon, \Delta\eta), \tag{51}$$

where $\|o(\Delta\varepsilon, \Delta\eta)\|$ denotes the higher-order remainder term satisfying $\|o(\Delta\varepsilon, \Delta\eta)\| = o(\|\Delta\varepsilon\| + \|\Delta\eta\|)$.

By (27),

$$\begin{aligned} S(\bar\varepsilon + \Delta\varepsilon) &= \{\eta \mid f(\bar\varepsilon + \Delta\varepsilon, \eta) + N_E(\eta) \ni 0\} \\ &= \{\eta \mid f(\bar\varepsilon, \bar\eta) + \nabla_\varepsilon f(\bar\varepsilon, \bar\eta)\Delta\varepsilon + \nabla_\eta f(\bar\varepsilon, \bar\eta)(\eta - \bar\eta) + o(\Delta\varepsilon, \eta - \bar\eta) + N_E(\eta) \ni 0\}. \end{aligned} \tag{52}$$

Then, $\eta \in S(\bar\varepsilon + \Delta\varepsilon)$ satisfieså that

$$f(\bar\varepsilon, \bar\eta) + \nabla_\varepsilon f(\bar\varepsilon, \bar\eta)\,\Delta\varepsilon + \nabla_\eta f(\bar\varepsilon, \bar\eta)(\eta - \bar\eta) + o(\Delta\varepsilon, \eta - \bar\eta) + N_E(\eta) \ni 0. \tag{53}$$

Rearranging the terms, we obtain the equivalent inclusion

$$-\nabla_\varepsilon f(\bar\varepsilon, \bar\eta)\,\Delta\varepsilon - o(\Delta\varepsilon, \eta - \bar\eta) \in f(\bar\varepsilon, \bar\eta) + \nabla_\eta f(\bar\varepsilon, \bar\eta)(\eta - \bar\eta) + N_E(\eta) = G(\eta), \tag{54}$$

where $G(\eta) := \nabla_\eta f(\bar\varepsilon, \bar\eta)\,(\eta - \bar\eta) + f(\bar\varepsilon, \bar\eta) + N_E(\eta)$ denotes the linearized generalized equation in $\eta$.

Therefore, any solution $\eta \in S(\bar\varepsilon + \Delta\varepsilon)$ can be written as:

$$\eta = G^{-1}(-\nabla_\varepsilon f(\bar\varepsilon, \bar\eta)\,\Delta\varepsilon - o(\Delta\varepsilon, \eta - \bar\eta)), \tag{55}$$

which expresses $S(\bar\varepsilon + \Delta\varepsilon)$ implicitly via the inverse mapping $G^{-1}$.

We use $\kappa$ to denote the Lipschitz constant of $G^{-1}$. Since $\bar\eta = G(0)$, we have:

we have

$$\begin{aligned} \|\eta - \bar\eta\| &= \|G^{-1}(-\nabla_\varepsilon f(\bar\varepsilon, \bar\eta)\,\Delta\varepsilon - o(\Delta\varepsilon, \eta - \bar\eta)) - G^{-1}(0)\| \\ &\leq k\,\| -\nabla_\varepsilon f(\bar\varepsilon, \bar\eta)\,\Delta\varepsilon - o(\Delta\varepsilon, \eta - \bar\eta)\|. \end{aligned} \tag{56}$$

Applying the triangle inequality gives

$$\|\eta - \bar\eta\| \leq k\,\|\nabla_\varepsilon f(\bar\varepsilon, \bar\eta)\|\,\|\Delta\varepsilon\| + k\,\|o(\Delta\varepsilon, \eta - \bar\eta)\|. \tag{57}$$

Since $o(\Delta\varepsilon, \eta - \bar\eta) = o(\|\Delta\varepsilon\| + \|\eta - \bar\eta\|)$ as $(\Delta\varepsilon, \eta - \bar\eta) \to (0,0)$, for any $\delta > 0$ there exists a neighborhood of $(0,0)$ such that

$$\|o(\Delta\varepsilon, \eta - \bar\eta)\| \leq \delta\big(\|\Delta\varepsilon\| + \|\eta - \bar\eta\|\big). \tag{58}$$

Combining (56) and (58) yields

$$\|\eta - \bar\eta\| \leq k\,\|\nabla_\varepsilon f(\bar\varepsilon, \bar\eta)\|\,\|\Delta\varepsilon\| + k\delta\,\|\Delta\varepsilon\| + k\delta\,\|\eta - \bar\eta\|. \tag{59}$$

Rearranging terms gives

$$(1 - k\delta)\,\|\eta - \bar\eta\| \leq k\big(\|\nabla_\varepsilon f(\bar\varepsilon, \bar\eta)\| + \delta\big)\,\|\Delta\varepsilon\|. \tag{60}$$

Finally, choosing $\delta > 0$ small enough such that $k\delta < \frac{1}{2}$, we obtain

$$\|\eta - \bar\eta\| \leq \frac{k}{1 - k\delta}\big(\|\nabla_\varepsilon f(\bar\varepsilon, \bar\eta)\| + \delta\big)\,\|\Delta\varepsilon\| = O(\|\Delta\varepsilon\|), \tag{61}$$

Since $G^{-1}$ is Lipschitz continuous,

$$S(\bar\varepsilon + \Delta\varepsilon) - \sigma(\bar\varepsilon + \Delta\varepsilon) = \big\|G^{-1}(-\nabla_\varepsilon f(\bar\varepsilon, \bar\eta)\Delta\varepsilon - o(\Delta\varepsilon, \eta - \bar\eta)) - G^{-1}(-\nabla_\varepsilon f(\bar\varepsilon, \bar\eta)\Delta\varepsilon)\big\|$$
$$\leq k\|o(\Delta\varepsilon, \eta - \bar\eta)\|. \tag{62}$$

Substituting (61) to (63) yields:

$$S(\bar\varepsilon + \Delta\varepsilon) - \sigma(\bar\varepsilon + \Delta\varepsilon) \leq k\|o(\Delta\varepsilon, \eta - \bar\eta)\| = o(\|\Delta\varepsilon\|). \tag{63}$$

This proves that $\sigma(\varepsilon)$ is a first-order approximation of $S(\varepsilon)$ at $\bar\varepsilon$. $\qquad\square$

### C.2 PROOF OF THEOREM 8

Following Lemma 19 and Remark 20, we establish that the mapping $G_0^{-1}$ is locally Lipschitz continuous. Moreover, by Lemma 21, it holds that

$$G^{-1}(v) = G_0^{-1}(v) + \bar\eta$$

which implies that $G^{-1}$ inherits the Lipschitz continuity and single-valuedness of $G_0^{-1}$. In this case, the Lipschitz continuity of $G^{-1}$ ensures the applicability of Lemma 22, which shows that $\sigma(\varepsilon)$ serves as a first-order local approximation of the solution mapping $S(\varepsilon)$..

Indeed, for small $\Delta\varepsilon$,

$$\sigma(\bar\varepsilon + \Delta\bar\varepsilon) = G^{-1}\big(-\nabla_\varepsilon f(\bar\varepsilon, \bar\eta)\,\Delta\bar\varepsilon\big) \tag{64}$$

$$= G_0^{-1}\big(-\nabla_\varepsilon f(\bar\varepsilon, \bar\eta)\,\Delta\bar\varepsilon\big) + \bar\eta. \tag{65}$$

Therefore,

$$S(\bar\varepsilon + \Delta\bar\varepsilon) - S(\bar\varepsilon) = \sigma(\bar\varepsilon + \Delta\bar\varepsilon) - \bar\eta + o(\|\Delta\varepsilon\|) \tag{66}$$
$$= G_0^{-1}\big(-\nabla_\varepsilon f(\bar\varepsilon, \bar\eta)\,\Delta\bar\varepsilon\big) + o(\|\Delta\bar\varepsilon\|)$$
$$= \bar s\big(-\nabla_\varepsilon f(\bar\varepsilon, \bar\eta)\,\Delta\bar\varepsilon\big) + o(\|\Delta\bar\varepsilon\|). \tag{67}$$

We obtain

$$\lim_{t\downarrow 0} \frac{S(\bar\varepsilon + t\Delta\bar\varepsilon) - S(\bar\varepsilon)}{t} = \lim_{t\downarrow 0} \frac{\bar s\big(-\nabla_\varepsilon f(\bar\varepsilon, \bar\eta)\,t\Delta\bar\varepsilon\big) + o(\|\Delta\bar\varepsilon\|)}{t} \tag{68}$$

$$\tag{69}$$

Since $N_K$ is the normal cone mapping of the convex cone $K$, it satisfies $N_K(\alpha w) = \alpha N_K(w)$ for all $\alpha > 0$. Together with the linearity of $A$, this gives $(A + N_K)(\alpha w) = \alpha (A + N_K)(w)$. Hence $\bar{s} = (A + N_K)^{-1}$ is positively homogeneous,

$$\lim_{t \downarrow 0} \frac{\bar{s}\big( - \nabla_\varepsilon f(\bar{\varepsilon}, \bar{\eta}) \, t \Delta \bar{\varepsilon}\big) + o(\|\Delta \bar{\varepsilon}\|)}{t} = \bar{s}(-\nabla_\varepsilon f(\bar{\varepsilon}, \bar{\eta}) \, \Delta \bar{\varepsilon}). \tag{70}$$

Hence, $S$ is directionally differentiable at $\bar{\varepsilon}$ with

$$DS(\bar{\varepsilon})(\Delta \bar{\varepsilon}) = \bar{s}\big( - \nabla_\varepsilon f(\bar{\varepsilon}, \bar{\eta}) \, \Delta \bar{\varepsilon}\big).$$

Equation (33) ensures that

$$\bar{s}\left(-\nabla_\varepsilon f(\bar{\varepsilon}, \bar{\eta}) \Delta \bar{\varepsilon}\right)$$

is the solution of the auxiliary VI (15). This implies that, under the assumptions of Theorem 8, the solution mapping $S$ is directionally differentiable at $\bar{\varepsilon}$, and its directional derivative is given by the solution of the auxiliary VI (15). Since $S(\varepsilon) = (\theta, \lambda)$, the DIF $D\theta(\bar{\varepsilon}; \Delta\bar{\varepsilon})$ defined in Definition 2 corresponds to the $\theta$-component of $DS(\bar{\varepsilon})(\Delta \bar{\varepsilon})$. This guarantees the existence of the DIF.

### C.3 PROOF OF PROPOSITION 5

*Proof.* Given $(\bar{\varepsilon}, \bar{\theta}, \bar{\lambda})$ satisfy the optimality condition (11), we have

$$\nabla_\theta L(\bar{\varepsilon}, \bar{\theta}, \bar{\lambda}) + N_{\mathbb{R}^d}(\bar{\theta}) \ni 0, \quad -\nabla_\lambda L(\bar{\varepsilon}, \bar{\theta}, \bar{\lambda}) + N_{\mathbb{R}^m_+}(\bar{\lambda}) \ni 0. \tag{71}$$

Following the definition of the normal cone (Def. 4), condition (71) implies that:

$$\theta \in \mathbb{R}^d, \nabla_\theta L(\bar{\varepsilon}, \bar{\theta}, \bar{\lambda}) = 0, \tag{72}$$

$$\lambda \in \mathbb{R}^m_+, -\nabla_\lambda L(\bar{\varepsilon}, \bar{\theta}, \bar{\lambda}) \leq 0. \tag{73}$$

Note that $R^m_+$ is a closed, convex cone. Following condition (21), we have

$$\nabla_\lambda L(\bar{\varepsilon}, \bar{\theta}, \bar{\lambda}) \cdot \bar{\lambda} = 0, \tag{74}$$

which is consistent with the complementary slackness condition in the KKT system.

Now we derive the equivalent of VI (13) and VI (15). We assume the $\Delta\bar{\varepsilon}, \Delta\hat{\theta}, \Delta\hat{\lambda}$ are sufficiently close to 0.

VI (13) requires that

$$\nabla_\theta L(\bar{\varepsilon}, \bar{\theta}, \bar{\lambda}) + \nabla_{\theta\varepsilon} L(\bar{\varepsilon}, \bar{\theta}, \bar{\lambda})\Delta\bar{\varepsilon} + \nabla^2_\theta L(\bar{\varepsilon}, \bar{\theta}, \bar{\lambda})\Delta\hat{\theta} + \nabla_{\theta\lambda} L(\bar{\varepsilon}, \bar{\theta}, \bar{\lambda})\Delta\hat{\lambda} + N_{\mathbb{R}^d}(\bar{\theta} + \Delta\hat{\theta}) \ni 0$$

$$\Updownarrow$$

$$\bar{\theta} + \Delta\hat{\theta} \in \mathbb{R}^d, \quad \nabla_\theta L(\bar{\varepsilon}, \bar{\theta}, \bar{\lambda}) + \nabla_{\theta\varepsilon} L(\bar{\varepsilon}, \bar{\theta}, \bar{\lambda})\Delta\bar{\varepsilon} + \nabla^2_\theta L(\bar{\varepsilon}, \bar{\theta}, \bar{\lambda})\Delta\hat{\theta} + \nabla_{\theta\lambda} L(\bar{\varepsilon}, \bar{\theta}, \bar{\lambda})\Delta\hat{\lambda} = 0 \tag{75}$$

and

$$- \nabla_\lambda L(\bar{\varepsilon}, \bar{\theta}, \bar{\lambda}) - \nabla_{\lambda\varepsilon} L(\bar{\varepsilon}, \bar{\theta}, \bar{\lambda})\Delta\bar{\varepsilon} - \nabla_{\lambda\theta} L(\bar{\varepsilon}, \bar{\theta}, \bar{\lambda})\Delta\hat{\theta} - \nabla^2_\lambda L(\bar{\varepsilon}, \bar{\theta}, \bar{\lambda})\Delta\hat{\lambda} + N_{\mathbb{R}^m_+}(\bar{\lambda} + \Delta\hat{\lambda}) \ni 0$$

$$\Updownarrow$$

$$\bar{\lambda} + \Delta\hat{\lambda} \in \mathbb{R}^m_+, \quad \nabla_\lambda L(\bar{\varepsilon}, \bar{\theta}, \bar{\lambda}) + \nabla_{\lambda\varepsilon} L(\bar{\varepsilon}, \bar{\theta}, \bar{\lambda})\Delta\bar{\varepsilon} + \nabla_{\lambda\theta} L(\bar{\varepsilon}, \bar{\theta}, \bar{\lambda})\Delta\hat{\theta} + \nabla^2_\lambda L(\bar{\varepsilon}, \bar{\theta}, \bar{\lambda})\Delta\hat{\lambda} \leq 0 \tag{76}$$

It is straightforward to verify that $\nabla^2_\lambda L(\bar{\varepsilon}, \bar{\theta}, \bar{\lambda}) = 0$. By equation (72), $\nabla_\theta L(\bar{\varepsilon}, \bar{\theta}, \bar{\lambda}) = 0$. Substituting $\nabla_\theta L(\bar{\varepsilon}, \bar{\theta}, \bar{\lambda}) = 0$ to (75) and $\nabla^2_\lambda L(\bar{\varepsilon}, \bar{\theta}, \bar{\lambda}) = 0$ to (76) yields

$$\bar{\theta} + \Delta\hat{\theta} \in \mathbb{R}^d, \quad \nabla_{\theta\varepsilon} L(\bar{\varepsilon}, \bar{\theta}, \bar{\lambda})\Delta\bar{\varepsilon} + \nabla^2_\theta L(\bar{\varepsilon}, \bar{\theta}, \bar{\lambda})\Delta\hat{\theta} + \nabla_{\theta\lambda} L(\bar{\varepsilon}, \bar{\theta}, \bar{\lambda})\Delta\hat{\lambda} = 0, \tag{77}$$

and

$$\bar{\lambda} + \Delta\hat{\lambda} \in \mathbb{R}_+^m, \quad \nabla_\lambda L(\bar{\varepsilon}, \bar{\theta}, \bar{\lambda}) + \nabla_{\lambda\varepsilon} L(\bar{\varepsilon}, \bar{\theta}, \bar{\lambda})\Delta\bar{\varepsilon} + \nabla_{\lambda\theta} L(\bar{\varepsilon}, \bar{\theta}, \bar{\lambda})\Delta\hat{\theta} \leq 0 \qquad (78)$$

Since $\mathbb{R}_+^m$ is a closed convex cone, by (21), we have

$$\left( \nabla_\lambda L(\bar{\varepsilon}, \bar{\theta}, \bar{\lambda}) + \nabla_{\lambda\varepsilon} L(\bar{\varepsilon}, \bar{\theta}, \bar{\lambda})\,\Delta\bar{\varepsilon} + \nabla_{\lambda\theta} L(\bar{\varepsilon}, \bar{\theta}, \bar{\lambda})\,\Delta\hat{\theta} \right) \cdot (\bar{\lambda} + \Delta\hat{\lambda}) = 0 \qquad (79)$$

In summary, VI (13) is equivalent to the system consisting of (77), (78), and (79). Note that $\lambda = [\lambda_1, \ldots, \lambda_j]$, we now further discuss the formulation (78) and (79) for $j \in I_{\text{Inactive}}, j \in I_{\text{Binding}}$, and $j \in I_{\text{Non-binding}}$.

**Case 1.** If $j \in I_{\text{Inactive}}$, then $\nabla_{\lambda_j} L(\bar{\varepsilon}, \bar{\theta}, \bar{\lambda}) < 0$ and $\bar{\lambda}_j = 0$. Since $\Delta\bar{\varepsilon}, \Delta\hat{\theta}$, and $\Delta\hat{\lambda}$ are all sufficiently close to 0 and $\nabla_{\lambda_j} L(\bar{\varepsilon}, \bar{\theta}, \bar{\lambda}) < 0$, condition (78)

$$\nabla_{\lambda_j} L(\bar{\varepsilon}, \bar{\theta}, \bar{\lambda}) + \nabla_{\lambda_j\varepsilon} L(\bar{\varepsilon}, \bar{\theta}, \bar{\lambda})\Delta\bar{\varepsilon} + \nabla_{\lambda_j\theta} L(\bar{\varepsilon}, \bar{\theta}, \bar{\lambda})\Delta\hat{\theta} \leq 0$$

is automatically satisfied. Therefore, the term

$$\nabla_{\lambda_j\varepsilon} L(\bar{\varepsilon}, \bar{\theta}, \bar{\lambda})\,\Delta\bar{\varepsilon} + \nabla_{\lambda_j\theta} L(\bar{\varepsilon}, \bar{\theta}, \bar{\lambda})\,\Delta\hat{\theta} \text{ is free.} \qquad (80)$$

Moreover, condition (79) implies that

$$\Delta\hat{\lambda}_j = 0 \quad \text{for } j \in I_{\text{Inactive}}. \qquad (81)$$

**Case 2.** If $j \in I_{\text{Non-binding}}$, then $\nabla_{\lambda_j} L(\bar{\varepsilon}, \bar{\theta}, \bar{\lambda}) = 0$ and $\bar{\lambda}_j = 0$. Substituting $\nabla_{\lambda_j} L(\bar{\varepsilon}, \bar{\theta}, \bar{\lambda}) = 0$ and $\bar{\lambda}_j = 0$ into condition (78) yields

$$\begin{aligned} \Delta\hat{\lambda}_j \geq 0 \qquad &\text{for } j \in I_{\text{Non-binding}}, \\ \nabla_{\lambda_j\varepsilon} L(\bar{\varepsilon}, \bar{\theta}, \bar{\lambda})\,\Delta\bar{\varepsilon} + \nabla_{\lambda_j\theta} L(\bar{\varepsilon}, \bar{\theta}, \bar{\lambda})\,\Delta\hat{\theta} \leq 0 \qquad &\text{for } j \in I_{\text{Non-binding}}. \end{aligned} \qquad (82)$$

**Case 3.** If $j \in I_{\text{Binding}}$, then $\nabla_{\lambda_j} L(\bar{\varepsilon}, \bar{\theta}, \bar{\lambda}) = 0$ and $\bar{\lambda}_j \geq 0$. Since $\Delta\hat{\lambda}$ is close to 0, $\bar{\lambda} + \Delta\hat{\lambda} \in \mathbb{R}_+^m$ is automatically satisfied. To satisfy conditions (78) and (79), we must have

$$\nabla_{\lambda_j\varepsilon} L(\bar{\varepsilon}, \bar{\theta}, \bar{\lambda})\,\Delta\bar{\varepsilon} + \nabla_{\lambda_j\theta} L(\bar{\varepsilon}, \bar{\theta}, \bar{\lambda})\,\Delta\hat{\theta} = 0.$$

In summary, we obtain

$$\begin{aligned} \Delta\hat{\lambda}_j \text{ is free.} \qquad &\text{for } j \in I_{\text{Binding}}, \\ \nabla_{\lambda_j\varepsilon} L(\bar{\varepsilon}, \bar{\theta}, \bar{\lambda})\,\Delta\bar{\varepsilon} + \nabla_{\lambda_j\theta} L(\bar{\varepsilon}, \bar{\theta}, \bar{\lambda})\,\Delta\hat{\theta} = 0 \qquad &\text{for } j \in I_{\text{Binding}}. \end{aligned} \qquad (83)$$

Taken together, Cases 1–3 show that VI (13) is equivalent to the system consisting of (77) and (80–83). On the other hand, the following argument shows that VI (15) is also equivalent to this system.

The VI (15) implies that:

$$\nabla_{\theta\varepsilon} L(\bar{\varepsilon}, \bar{\theta}, \bar{\lambda})\Delta\bar{\varepsilon} + \nabla_\theta^2 L(\bar{\varepsilon}, \bar{\theta}, \bar{\lambda})\Delta\hat{\theta} + \nabla_{\theta\lambda} L(\bar{\varepsilon}, \bar{\theta}, \bar{\lambda})\Delta\hat{\lambda} + N_{\mathbb{R}^d}(\Delta\hat{\theta}) \ni 0$$

$$\Updownarrow \qquad\qquad\qquad\qquad\qquad\qquad\qquad\qquad\qquad (84)$$

$$\Delta\hat{\theta} \in \mathbb{R}^d, \quad \nabla_{\theta\varepsilon} L(\bar{\varepsilon}, \bar{\theta}, \bar{\lambda})\Delta\bar{\varepsilon} + \nabla_\theta^2 L(\bar{\varepsilon}, \bar{\theta}, \bar{\lambda})\Delta\hat{\theta} + \nabla_{\theta\lambda} L(\bar{\varepsilon}, \bar{\theta}, \bar{\lambda})\Delta\hat{\lambda} = 0$$

and

$$-\nabla_{\lambda\varepsilon} L(\bar{\varepsilon}, \bar{\theta}, \bar{\lambda})\Delta\bar{\varepsilon} - \nabla_{\lambda\theta} L(\bar{\varepsilon}, \bar{\theta}, \bar{\lambda})\Delta\hat{\theta} - \nabla_\lambda^2 L(\bar{\varepsilon}, \bar{\theta}, \bar{\lambda})\Delta\hat{\lambda} + N_D(\Delta\hat{\lambda}) \ni 0$$

$$\Updownarrow$$

$$\Delta\hat{\lambda} \in D, \quad (-\nabla_{\lambda\varepsilon} L(\bar{\varepsilon}, \bar{\theta}, \bar{\lambda})\Delta\bar{\varepsilon} - \nabla_{\lambda\theta} L(\bar{\varepsilon}, \bar{\theta}, \bar{\lambda})\Delta\hat{\theta} - \nabla_\lambda^2 L(\bar{\varepsilon}, \bar{\theta}, \bar{\lambda})\Delta\hat{\lambda})(\Delta\hat{\lambda}' - \Delta\hat{\lambda}) \leq 0, \forall \Delta\hat{\lambda}' \in D$$

$$(85)$$

Since $D$ is a closed convex cone, by (21), we have

$$\left(\nabla_{\lambda\varepsilon}L(\bar{\varepsilon},\bar{\theta},\bar{\lambda})\,\Delta\bar{\varepsilon}+\nabla_{\lambda\theta}L(\bar{\varepsilon},\bar{\theta},\bar{\lambda})\,\Delta\hat{\theta}\right)\cdot\Delta\hat{\lambda}=0 \tag{86}$$

**Case 1.** If $j\in I_{\text{Inactive}}$, then by the definition of the space $D$ (see Proposition 5), we have

$$\Delta\hat{\lambda}_j=0\ ,\forall\Delta\hat{\lambda}\in D \tag{87}$$

Thus, condition (85) is automatically satisfied. Therefore, the term

$$\nabla_{\lambda_j\varepsilon}L(\bar{\varepsilon},\bar{\theta},\bar{\lambda})\,\Delta\bar{\varepsilon}+\nabla_{\lambda_j\theta}L(\bar{\varepsilon},\bar{\theta},\bar{\lambda})\,\Delta\hat{\theta}\ \text{ is free.} \tag{88}$$

**Case 2.** If $j\in I_{\text{Non-binding}}$, then by the definition of the space $D$ (see Proposition 5), we have

$$\Delta\hat{\lambda}_j\geq 0,\ \ \forall\Delta\hat{\lambda}\in D \tag{89}$$

By the condition (86),

$$\nabla_{\lambda_j\varepsilon}L(\bar{\varepsilon},\bar{\theta},\bar{\lambda})\Delta\bar{\varepsilon}+\nabla_{\lambda_j\theta}L(\bar{\varepsilon},\bar{\theta},\bar{\lambda})\Delta\hat{\theta}\leq 0\ \ \ \text{ for }j\in I_{\text{Non-binding}}\,. \tag{90}$$

**Case 3.** If $j\in I_{\text{binding}}$, then by the definition of the space $D$ (see Proposition 5), we have

$$\Delta\hat{\lambda}_j\ \ \text{ is free} \tag{91}$$

By the condition (86),

$$\nabla_{\lambda_j\varepsilon}L(\bar{\varepsilon},\bar{\theta},\bar{\lambda})\Delta\bar{\varepsilon}+\nabla_{\lambda_j\theta}L(\bar{\varepsilon},\bar{\theta},\bar{\lambda})\Delta\hat{\theta}=0\ \ \ \text{ for }j\in I_{\text{Binding}} \tag{92}$$

Since (84,87–92) and (77,80–83) are equivalent formulations, VI (15) is equivalent to VI (13). $\quad\square$
$\square$

## C.4 Proof of corollary 3

*Proof.* Recall the directional derivative

$$D\theta(\bar{\varepsilon};v):=\lim_{t\downarrow 0,v'\to v}\frac{\theta\left(\bar{\varepsilon}+tv'\right)-\theta(\bar{\varepsilon})}{t},$$

whenever the limit exists and is finite.

Let $\alpha\geq 0$. - If $\alpha=0$ : by the definition with $v\equiv 0$,

$$D\theta(\bar{\varepsilon};0)=\lim_{t\downarrow 0,v'\to 0}\frac{\theta\left(\bar{\varepsilon}+tv'\right)-\theta(\bar{\varepsilon})}{t}=0,$$

hence $0\cdot D\theta(\bar{\varepsilon};v)=D\theta(\bar{\varepsilon};0)$. - If $\alpha>0$ : using the definition with the direction $\alpha v$,

$$D\theta(\bar{\varepsilon};\alpha v)=\lim_{t\downarrow 0,u'\to\alpha v}\frac{\theta\left(\bar{\varepsilon}+tu'\right)-\theta(\bar{\varepsilon})}{t}.$$

Choose $u'=\alpha v'$ with $v'\to v$. Then

$$D\theta(\bar{\varepsilon};\alpha v)=\lim_{t\downarrow 0,v'\to v}\frac{\theta\left(\bar{\varepsilon}+t\left(\alpha v'\right)\right)-\theta(\bar{\varepsilon})}{t}=\lim_{t\downarrow 0,v'\to v}\frac{\theta\left(\bar{\varepsilon}+(\alpha t)v'\right)-\theta(\bar{\varepsilon})}{t}$$

$$=\lim_{s\downarrow 0,v'\to v}\frac{\theta\left(\bar{\varepsilon}+sv'\right)-\theta(\bar{\varepsilon})}{s/\alpha}\ \ \ (\text{ set }s=\alpha t)$$

$$=\alpha\lim_{s\downarrow 0,v'\to v}\frac{\theta\left(\bar{\varepsilon}+sv'\right)-\theta(\bar{\varepsilon})}{s}=\alpha D\theta(\bar{\varepsilon};v).$$

Combining the two cases yields

$$\alpha D\theta(\bar{\varepsilon};\Delta\bar{\varepsilon})=D\theta(\bar{\varepsilon};\alpha\Delta\bar{\varepsilon}),\quad\forall\alpha\in\mathbb{R}^+$$

$\square$

## C.5 PROOF OF PROPOSITION 6

Section C.2 has proved that the directional derivative of the solution mapping $S$ is characterized by the auxiliary VI (15). Specifically,

$$DS(\bar{\varepsilon})(\Delta\bar{\varepsilon}) = \Delta\hat{\eta} \quad \text{where} \quad \mu\,\Delta\bar{\varepsilon} + A\,\Delta\hat{\eta} + N_K(\Delta\hat{\eta}) \ni 0. \tag{93}$$

Since the DIF $D\theta(\bar{\varepsilon}; \Delta\bar{\varepsilon})$ defined in Definition 2 is the $\theta$-component of $DS(\bar{\varepsilon})(\Delta\bar{\varepsilon})$, we have

$$\Delta\hat{\theta} = D\theta(\bar{\varepsilon}; \Delta\bar{\varepsilon}). \tag{94}$$

## C.6 PROOF OF PROPOSITION 9

*Proof.* Let $\Delta\hat{\eta} = [\Delta\hat{\theta}, \Delta\hat{\lambda}]$ denote the solution of Auxiliary VI (15). By (33),

$$\Delta\hat{\eta} = \bar{s}(-u\,\Delta\bar{\varepsilon}).$$

By definition,

$$\Delta\eta = \eta - \bar{\eta} = S(\bar{\varepsilon} + \Delta\bar{\varepsilon}) - \bar{\eta}.$$

Therefore,

$$\|\Delta\theta - \Delta\hat{\theta}\| \le \|\Delta\eta - \Delta\hat{\eta}\| = \big\|S(\bar{\varepsilon} + \Delta\bar{\varepsilon}) - \bar{\eta} - \bar{s}(-u\,\Delta\bar{\varepsilon})\big\|. \tag{95}$$

By Lemma 22,

$$\sigma(\varepsilon) := G^{-1}\Big(-\nabla_\varepsilon f(\bar{\varepsilon}, \bar{\eta})\,(\varepsilon - \bar{\varepsilon})\Big)$$

is a first-order approximation of $S(\varepsilon)$ and $\sigma(\bar{\varepsilon}) = S(\bar{\varepsilon}) = \bar{\eta}$. Thus,

$$S(\bar{\varepsilon} + \Delta\bar{\varepsilon}) = \sigma(\bar{\varepsilon} + \Delta\bar{\varepsilon}) + o(\|\Delta\bar{\varepsilon}\|). \tag{96}$$

By Lemma 21,

$$\sigma(\bar{\varepsilon} + \Delta\bar{\varepsilon}) = G^{-1}(-u\,\Delta\bar{\varepsilon}) = G_0^{-1}(-u\,\Delta\bar{\varepsilon}) + \bar{\eta} = \bar{s}(-u\,\Delta\bar{\varepsilon}) + \bar{\eta}. \tag{97}$$

Substituting (97) and (96) into right side of (95) yields

$$\big\|S(\bar{\varepsilon} + \Delta\bar{\varepsilon}) - \bar{\eta} - \bar{s}(-u\,\Delta\bar{\varepsilon})\big\| = o(\|\Delta\bar{\varepsilon}\|),$$

hence

$$\|\Delta\theta - \Delta\hat{\theta}\| \le o(\|\Delta\bar{\varepsilon}\|).$$

By the little-$o$ definition, for every $\epsilon > 0$ there exists $\delta(\epsilon) > 0$ such that, whenever $0 < \|\Delta\bar{\varepsilon}\| < \delta(\epsilon)$,

$$\frac{\|\Delta\theta - \Delta\hat{\theta}\|}{\|\Delta\bar{\varepsilon}\|} < \epsilon. \tag{98}$$

Fix any $M > 0$ and set $\epsilon := M$. Then there exists $\delta := \delta(M) > 0$ such that, for all $0 < \|\Delta\bar{\varepsilon}\| < \delta$,

$$\|\Delta\theta - \Delta\hat{\theta}\| \le M\,\|\Delta\bar{\varepsilon}\|. \tag{99}$$

$\square$

## C.7 PROOF OF THEOREM 10

Define the Lagrangian of QP (18) as

$$\mathcal{L}_{QP}(\omega, \zeta) = L(\bar{\varepsilon}, \bar{\theta}, \bar{\lambda}) + \big\langle \nabla_{\theta\varepsilon}L(\bar{\varepsilon}, \bar{\theta}, \bar{\lambda})\cdot\Delta\bar{\varepsilon},\ \omega \big\rangle + \frac{1}{2}\big\langle \omega,\ \nabla^2_{\theta\theta}L(\bar{\varepsilon}, \bar{\theta}, \bar{\lambda})\,\omega \big\rangle$$

$$+ \sum_{j \in I_{\text{binding}}} \zeta_j \left[ \frac{1}{N_j}\sum_{i=1}^{N_j}\ell_j(z_i^{(j)}, \bar{\theta}) + \frac{1}{N_j}\sum_{i=1}^{N_j}\nabla_\theta\ell_j(z_i^{(j)}, \bar{\theta})\cdot\omega + \sum_{z_i^{(j)} \in Z^r}\Delta\bar{\varepsilon}_j\,\ell_j(z_i^{(j)}, \bar{\theta}) - \tau_j \right]$$

$$+ \sum_{j \in I_{\text{non-binding}}} \zeta_j \left[ \frac{1}{N_j}\sum_{i=1}^{N_j}\ell_j(z_i^{(j)}, \bar{\theta}) + \frac{1}{N_j}\sum_{i=1}^{N_j}\nabla_\theta\ell_j(z_i^{(j)}, \bar{\theta})\cdot\omega + \sum_{z_i^{(j)} \in Z^r}\Delta\bar{\varepsilon}_j\,\ell_j(z_i^{(j)}, \bar{\theta}) - \tau_j \right]$$

$$+ \sum_{j \in I_{\text{Inactive}}} \zeta_j \left[ \frac{1}{N_j}\sum_{i=1}^{N_j}\ell_j(z_i^{(j)}, \bar{\theta}) + \frac{1}{N_j}\sum_{i=1}^{N_j}\nabla_\theta\ell_j(z_i^{(j)}, \bar{\theta})\cdot\omega + \sum_{z_i^{(j)} \in Z^r}\Delta\bar{\varepsilon}_j\,\ell_j(z_i^{(j)}, \bar{\theta}) - \tau_j \right].$$

$$\tag{100}$$

A pair $(\omega^\star, \zeta^\star)$ satisfies the Karush–Kuhn–Tucker (KKT) conditions if:

(i) Primal feasibility.

$$\frac{1}{N_j}\sum_{i=1}^{N_j}\ell_j(z_i^{(j)}, \bar\theta) + \frac{1}{N_j}\sum_{i=1}^{N_j}\nabla_\theta\ell_j(z_i^{(j)}, \bar\theta)\cdot\omega^\star + \sum_{z_i^{(j)}\in Z^r}\Delta\bar\varepsilon_j\,\ell_j(z_i^{(j)}, \bar\theta) - \tau_j = 0, \qquad \forall j \in I_{\text{binding}},$$
(101)

$$\frac{1}{N_j}\sum_{i=1}^{N_j}\ell_j(z_i^{(j)}, \bar\theta) + \frac{1}{N_j}\sum_{i=1}^{N_j}\nabla_\theta\ell_j(z_i^{(j)}, \bar\theta)\cdot\omega^\star + \sum_{z_i^{(j)}\in Z^r}\Delta\bar\varepsilon_j\,\ell_j(z_i^{(j)}, \bar\theta) - \tau_j \le 0, \qquad \forall j \in I_{\text{non-binding}}.$$
(102)

For $j \in I_{\text{Inactive}}$, there is no constraint.

(ii) Stationarity.

$$\nabla_{\theta\varepsilon}L(\bar\varepsilon, \bar\theta, \bar\lambda)\cdot\Delta\bar\varepsilon + \nabla_{\theta\theta}^2 L(\bar\varepsilon, \bar\theta, \bar\lambda)\,\omega^\star + \sum_{j\in I_{\text{binding}}\cup I_{\text{non-binding}}\cup I_{\text{Inactive}}}\zeta_j^\star\left(\frac{1}{N_j}\sum_{i=1}^{N_j}\nabla_\theta\ell_j(z_i^{(j)}, \bar\theta)\right) = 0. \quad (103)$$

(iii) Dual feasibility.

$$\zeta_j^\star \ge 0, \qquad \forall j \in I_{\text{non-binding}}. \tag{104}$$

(iv) Complementary slackness.

$$\zeta_j^\star\left[\frac{1}{N_j}\sum_{i=1}^{N_j}\ell_j(z_i^{(j)}, \bar\theta) + \frac{1}{N_j}\sum_{i=1}^{N_j}\nabla_\theta\ell_j(z_i^{(j)}, \bar\theta)\cdot\omega^\star + \sum_{z_i^{(j)}\in Z^r}\Delta\bar\varepsilon_j\,\ell_j(z_i^{(j)}, \bar\theta) - \tau_j\right] = 0, \qquad \forall j \in I_{\text{non-binding}}.$$
(105)

We now show that any $(\omega^\star, \zeta^\star)$ satisfying the above KKT conditions also satisfies the auxiliary VI (15). Recall that in Section C.3, we have proved that the auxiliary VI (15) is equivalent to the system $(84, 87 - -92)$.

Note that

$$L(\varepsilon, \theta, \lambda) = \frac{1}{N_0}\sum_{i=1}^{N_0}\ell_0(z_i^{(0)}, \theta) + \varepsilon_0\sum_{z_i^{(0)}\in Z^r}\ell_0(z_i^{(0)}, \theta)$$

$$+ \sum_{j=1}^{m}\lambda_j\left[\frac{1}{N_j}\sum_{i=1}^{N_j}\ell_j(z_i^{(j)}, \theta) + \varepsilon_j\sum_{z_i^{(j)}\in Z^r}\ell_j(z_i^{(j)}, \theta) - \tau_j\right],$$

we have

$$\frac{1}{N_j}\sum_{i=1}^{N_j}\nabla_\theta\ell_j(z_i^{(j)}, \bar\theta) = \nabla_{\theta\lambda_j}L(\bar\varepsilon, \bar\theta, \bar\lambda). \tag{106}$$

The stationarity condition can be rewritten as

$$\nabla_{\theta\varepsilon}L(\bar\varepsilon, \bar\theta, \bar\lambda)\,\Delta\bar\varepsilon + \nabla_\theta^2 L(\bar\varepsilon, \bar\theta, \bar\lambda)\,\omega^\star + \nabla_{\theta\lambda}L(\bar\varepsilon, \bar\theta, \bar\lambda)\,\zeta^\star = 0. \tag{107}$$

By the definition of the normal cone (Definition 4), this is equivalent to

$$\nabla_{\theta\varepsilon}L(\bar\varepsilon, \bar\theta, \bar\lambda)\,\Delta\bar\varepsilon + \nabla_\theta^2 L(\bar\varepsilon, \bar\theta, \bar\lambda)\,\omega^\star + \nabla_{\theta\lambda}L(\bar\varepsilon, \bar\theta, \bar\lambda)\,\zeta^\star \in N_{\mathbb{R}^d}(\Delta\hat\theta). \tag{108}$$

We distinguish the three index sets.

**Case 1** ($j \in I_{\text{Inactive}}$)**.** By complementary slackness, $\zeta_j^\star = 0$.

**Case 2** ($j \in I_{\text{non-binding}}$). By complementary slackness, $\zeta_j^\star \geq 0$. Moreover, since $\frac{1}{N_j} \sum_{i=1}^{N_j} \ell_j(z_i^{(j)}, \bar{\theta}) = \tau_j$, substituting $\nabla_{\lambda_j \varepsilon} L(\bar{\varepsilon}, \bar{\theta}, \bar{\lambda}) \Delta \bar{\varepsilon} = \sum_{z_i^{(j)} \in Z^r} \Delta \bar{\varepsilon}_j \ell_j(z_i^{(j)}, \bar{\theta})$ and $\nabla_{\lambda_j \theta} L(\bar{\varepsilon}, \bar{\theta}, \bar{\lambda}) \omega^\star = \frac{1}{N_j} \sum_{i=1}^{N_j} \nabla_\theta \ell_j(z_i^{(j)}, \bar{\theta}) \omega^\star$ into the primal feasibility condition gives

$$\nabla_{\lambda_j \varepsilon} L(\bar{\varepsilon}, \bar{\theta}, \bar{\lambda}) \Delta \bar{\varepsilon} + \nabla_{\lambda_j \theta} L(\bar{\varepsilon}, \bar{\theta}, \bar{\lambda}) \omega^* \leq 0. \tag{109}$$

**Case 3** ($j \in I_{\text{binding}}$). Here $\zeta_j^\star$ is free. Since $\frac{1}{N_j} \sum_{i=1}^{N_j} \ell_j(z_i^{(j)}, \bar{\theta}) = \tau_j$, using the same substitutions into of case 3 into primal feasibility as above yields

$$\nabla_{\lambda_j \varepsilon} L(\bar{\varepsilon}, \bar{\theta}, \bar{\lambda}) \Delta \bar{\varepsilon} + \nabla_{\lambda_j \theta} L(\bar{\varepsilon}, \bar{\theta}, \bar{\lambda}) \omega^* = 0. \tag{110}$$

Combining all cases, $\zeta^\star$ satisfies

$$\nabla_{\lambda \varepsilon} L(\bar{\varepsilon}, \bar{\theta}, \bar{\lambda}) \Delta \bar{\varepsilon} - \nabla_{\lambda \theta} L(\bar{\varepsilon}, \bar{\theta}, \bar{\lambda}) \omega^\star - \nabla_\lambda^2 L(\bar{\varepsilon}, \bar{\theta}, \bar{\lambda}) \zeta^\star + N_{D'}(\zeta^\star) \ni 0, \tag{111}$$

where

$$D' := \left\{ \zeta^* \in \mathbb{R}^m \;\middle|\; \zeta_j^* \geq 0 \; (j \in I_{\text{non-binding}}), \; \zeta_j^* = 0 \; (j \in I_{\text{Inactive}}) \right\}.$$

(111) together with (107) shows that $(\omega^\star, \zeta^\star)$ satisfies

$$0 \in A(\omega^\star, \zeta^\star) + \mu \Delta \bar{\varepsilon} + N_K(\omega^\star, \zeta^\star),$$

which is exactly the auxiliary VI (15). $\qquad\square$

## C.8 PROOF OF PROPOSITION 12

*Proof.* By Theorem 10 we have $\omega^\star = \Delta \hat{\theta}$, where $\Delta \hat{\theta}$ is the $\theta$-component of the solution to the auxiliary VI (15). By Proposition 6, $\Delta \hat{\theta} = D\theta(\bar{\varepsilon}; \Delta \bar{\varepsilon})$. Therefore, $\omega^*$ is the DIF.

Assume that no constraints are active at $\bar{\theta}$. Then the KKT conditions reduce to the unconstrained optimality system, and the QP (18) has the unique solution $\omega^\star$ given by the Newton/IF direction:

$$\omega^\star = -\left[ \nabla_{\theta\theta}^2 L(\bar{\varepsilon}, \bar{\theta}, \bar{\lambda}) \right]^{-1} \nabla_{\theta\varepsilon} L(\bar{\varepsilon}, \bar{\theta}, \bar{\lambda}). \tag{112}$$

Since "no active constraints" implies $\bar{\lambda} = 0$, substituting $\bar{\lambda} = 0$ into (112) yields

$$\omega^\star = -\left[ \nabla_{\theta\theta}^2 L(\bar{\varepsilon}, \bar{\theta}, 0) \right]^{-1} \nabla_{\theta\varepsilon} L(\bar{\varepsilon}, \bar{\theta}, 0) = \left. \frac{d\theta(\varepsilon)}{d\varepsilon_0} \right|_{\varepsilon_0 = 0},$$

which is exactly the *influence function* (IF) defined in Koh & Liang (2017).

Therefore, in the inactive-constraint case, the QP solution $\omega^\star$ coincides with the IF, as claimed. $\quad\square$

## D DETAILS IN TOY EXAMPLE

We consider the toy example (3)

$$\min_{\theta \in \mathbb{R}^2} \frac{1}{3}(\theta^\top x_1 - y_1)^2 + \frac{1}{3}(\theta^\top x_2 - y_2)^2 + \frac{1}{3}(\theta^\top x_3 - y_3)^2 + \varepsilon(\theta^\top x_2 - y_2)^2 \quad \text{s.t.} \quad \|\theta\|_2^2 \leq 1,$$

with data $x_1 = (1, 0)$, $x_2 = (1, 0)$, $x_3 = (0, 1)$ and $y_1 = 1$, $y_2 = 0$, $y_3 = \frac{1}{2}$. At $\varepsilon = 0$, the constrained solution is $\bar{\theta} = (0.5, 0.5)$. The corresponding dual variable is $\bar{\lambda} = 0$.

**Gradients and Hessian.** For a single squared loss $\ell_i(\theta) = (\theta^\top x_i - y_i)^2$,

$$\nabla_\theta \ell_i(\theta) = 2(\theta^\top x_i - y_i) x_i, \qquad \nabla_{\theta\theta}^2 \ell_i(\theta) = 2 x_i x_i^\top.$$

Hence, at $\varepsilon = 0$ the (unconstrained) Hessian of $\frac{1}{3} \sum_{i=1}^3 \ell_i$ is

$$H = \frac{1}{3} \sum_{i=1}^3 2 x_i x_i^\top = \frac{2}{3}(x_1 x_1^\top + x_2 x_2^\top + x_3 x_3^\top) = \begin{pmatrix} \frac{4}{3} & 0 \\ 0 & \frac{2}{3} \end{pmatrix}. \tag{113}$$

The mixed derivative of the perturbed part is

$$\nabla_{\theta\varepsilon} L(\bar{\varepsilon}, \bar{\theta}) = \nabla_\theta \ell_2(\bar{\theta}) = 2(\bar{\theta}^\top x_2 - y_2) x_2 = 2 \cdot 0.5 \cdot (1, 0) = (1, 0). \tag{114}$$

### D.1 CLASSICAL IF (IGNORING CONSTRAINTS)

Differentiating the stationarity condition $\nabla_\theta L(\varepsilon, \theta) = 0$ w.r.t. $\varepsilon$ at $(\bar{\varepsilon}, \bar{\theta})$ gives

$$H \frac{d\theta}{d\varepsilon} + \nabla_{\theta\varepsilon} L(\bar{\varepsilon}, \bar{\theta}) = 0 \quad \Rightarrow \quad \frac{d\theta}{d\varepsilon} = -H^{-1} \nabla_{\theta\varepsilon} L(\bar{\varepsilon}, \bar{\theta}).$$

Using equation 113–equation 114,

$$\frac{d\theta}{d\varepsilon} = - \begin{pmatrix} \frac{3}{4} & 0 \\ 0 & \frac{3}{2} \end{pmatrix} (1, 0) = \left( -\frac{3}{4}, 0 \right).$$

Removing sample $z_2$ corresponds to $\Delta\varepsilon = -\frac{1}{3}$, so the IF estimate is

$$\Delta\theta_{\text{IF}} \approx \frac{d\theta}{d\varepsilon} \Delta\varepsilon = \left( -\frac{3}{4}, 0 \right) \cdot \left( -\frac{1}{3} \right) = \left( \frac{1}{4}, 0 \right). \tag{115}$$

Then $\bar{\theta} + \Delta\theta_{\text{IF}} = (0.75, 0.5)$ whose $\ell_1$-norm equals 1.25, i.e., the IF step is infeasible.

### D.2 DIF (FEASIBLE, SENSITIVITY ON THE LINEARIZED VI)

DIF linearizes the KKT/VI system at $(\bar{\varepsilon}, \bar{\theta})$ and searches $\Delta\theta$ in the tangent subspace. The QP (18) corresponding to the toy problem (3) is

$$\min_{\Delta\theta \in \mathbb{R}^2} \ \left\langle \nabla_{\theta\varepsilon} L(\bar{\varepsilon}, \bar{\theta}) \Delta\varepsilon, \ \Delta\theta \right\rangle + \frac{1}{2} \Delta\theta^\top H \Delta\theta \quad \text{s.t.} \quad \Delta\theta_1 + \Delta\theta_2 = 0. \tag{116}$$

With $\Delta\varepsilon = -\frac{1}{3}$ and $b := \nabla_{\theta\varepsilon} L \Delta\varepsilon = (-\frac{1}{3}, 0)$, (116) is

$$\min_{\Delta\theta} \ b^\top \Delta\theta + \frac{1}{2} \Delta\theta^\top H \Delta\theta \quad \text{s.t.} \quad \Delta\theta_1 + \Delta\theta_2 = 0.$$

The KKT conditions (with multiplier $\mu$) are

$$H\Delta\theta + b + \mu(1,1)^\top = 0, \qquad \Delta\theta_1 + \Delta\theta_2 = 0.$$

Expanding coordinates yields

$$\frac{4}{3}\Delta\theta_1 - \frac{1}{3} + \mu = 0, \qquad \frac{2}{3}\Delta\theta_2 + \mu = 0, \qquad \Delta\theta_1 + \Delta\theta_2 = 0.$$

From the second and third equations $\mu = -\frac{2}{3}\Delta\theta_2$ and $\Delta\theta_2 = -\Delta\theta_1$. Substituting into the first gives $\frac{4}{3}\Delta\theta_1 - \frac{1}{3} - \frac{2}{3}\Delta\theta_2 = 0 \Rightarrow 6\Delta\theta_1 = 1$, hence

$$\Delta\theta_{\text{DIF}} = (\tfrac{1}{6}, -\tfrac{1}{6}) \tag{117}$$

which lies on the tangent subspace and thus closely preserves feasibility to first order.

## E DETAILS IN CONSTRAINED LINEAR REGRESSION

We consider per-sample

$$\ell_i(\theta) = \tfrac{1}{2} (x_i^\top \theta - y_i)^2,$$

The constrained optimum $\bar{\theta}$ solves

$$\min_\theta \ \frac{1}{n} \sum_{i=1}^n \ell_i(\theta) \tag{118}$$

$$\text{s.t.} \quad A_{\text{eq}}\theta = b_{\text{eq}}, \quad A_{\text{ineq}}\theta \leq b_{\text{ineq}}.$$

Assume we remove the data sample $(x_i, y_i)$. We use

$$H = \nabla^2 L(\bar{\theta}) = \frac{1}{n} X^\top X, \qquad g_i = \nabla_\theta \ell_i(\bar{\theta}) = (x_i^\top \bar{\theta} - y_i) x_i.$$

**Classical Influence Function (IF).** Ignoring feasibility constraints, the first-order effect of removing sample $(x_i, y_i)$ is

$$\Delta\theta_{\text{IF}} = -\frac{1}{n} H^{-1} g_i. \tag{119}$$

**Penalty-based IF.** We approximate feasibility via a penalized surrogate

$$\tilde{L}(\theta) = \frac{1}{2n}\|X\theta - y\|_2^2 + \frac{\rho}{2}\|A_{\mathrm{eq}}\theta - b_{\mathrm{eq}}\|_2^2 + \sum_{j=1}^m k\left((a_j^\top \theta - b_j)_+\right)^3, \tag{120}$$

where $(t)_+ = \max\{t, 0\}$, $\rho, k > 0$, and $a_j^\top$ is the $j$-th row of $A_{\mathrm{ineq}}$. The Hessian at $\bar{\theta}$ is

$$H_{\mathrm{pen}} = \frac{1}{n}X^\top X + \rho\,A_{\mathrm{eq}}^\top A_{\mathrm{eq}} + \sum_{j\in\mathcal{A}} 6k\,t_j\,a_ja_j^\top, \qquad t_j = a_j^\top\bar{\theta} - b_j, \quad \mathcal{A} = \{j: t_j > 0\}. \tag{121}$$

The corresponding update is

$$\Delta\theta_{\mathrm{penIF}} = -\frac{1}{n}H_{\mathrm{pen}}^{-1}\,g_i. \tag{122}$$

In the experiments shown in Figure 3, we set the penalty parameters to $\rho = 200$ and $k = 1000$. To justify this choice, we evaluated the discrepancy between Penalty-based IF estimation $\Delta\theta_{\mathrm{penIF}}$ and leave-one-out retraining $\Delta\theta_{\mathrm{LOO}}$:

$$\|\Delta\theta_{\mathrm{penIF}} - \Delta\theta_{\mathrm{LOO}}\|.$$

by varying both $k$ and $\rho$.

Recall that $\bar{\theta}$ denotes the optimum of the constrained linear regression problem (118). At $\bar{\theta}$, all inequality constraints are strictly satisfied (i.e., $a_j^\top\bar{\theta} - b_j \leq 0$ for every $j$), which implies that the active set is empty: $\mathcal{A} = \varnothing$. Consequently, the cubic penalty term contributes no curvature to the local Hessian $H_{\mathrm{pen}}$, and therefore the value of $k$ does not influence the penalty-based IF update $\Delta\theta_{\mathrm{penIF}}$. This behavior is precisely reflected in Figure 6(b), where varying $k$ produces essentially identical estimates.

In contrast, Figure 6(a) shows that increasing the equality–constraint penalty $\rho$ steadily improves the approximation accuracy. The improvement becomes marginal once $\rho \geq 10$, after which the curve nearly plateaus. Based on this behavior, we adopt $\rho = 200$ in our experiments to ensure both stability and sufficient accuracy.

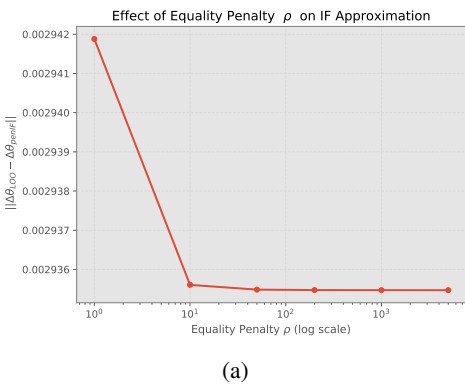
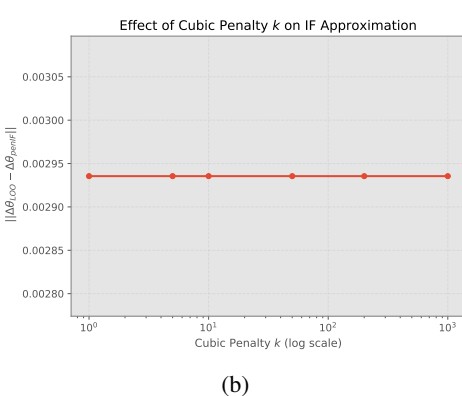

(a)                                    (b)

Figure 6: Hyper-parameter sensitivity of the penalty-based influence function. (a) Effect of the equality penalty parameter $\rho$ on the approximation error $\|\Delta\theta_{\mathrm{penIF}} - \Delta\theta_{\mathrm{LOO}}\|$. Increasing $\rho$ improves the accuracy, but the gain becomes negligible once $\rho \geq 10$. (b) Effect of the cubic penalty parameter $k$ on the cosine similarity between $\Delta\theta_{\mathrm{penIF}}$ and $\Delta\theta_{\mathrm{LOO}}$. Since the inequality constraint is non-active at the optimum $\bar{\theta}$, varying $k$ results in almost no change.

**Directional Influence Function (DIF).** We enforce feasibility with a one-step constrained QP (linearized KKT):

$$\Delta\theta_{\mathrm{DIF}} = \arg\min_{\Delta\theta\in\mathbb{R}^d} \quad \frac{1}{2}\Delta\theta^\top H\,\Delta\theta - \frac{1}{n}g_i^\top\Delta\theta$$
$$\text{s.t.} \quad A_{\mathrm{eq}}\Delta\theta = 0, \tag{123}$$
$$A_{\mathrm{ineq}}\Delta\theta = 0 \;\;(\text{active constraints only}).$$

# F   Primal–Dual Interior-Point Method for Solving the DIF Quadratic Program

We recall that the quadratic model (18) used in DIF takes the following form:

$$\min_{\omega} \quad b^\top \omega + \frac{1}{2}\, \omega^\top H_{\theta\theta} \omega \tag{124}$$

$$\text{s.t.} \quad g_j(\omega) := a_j^\top \omega + \eta_j \begin{cases} = 0, & j \in I_{\text{binding}}, \\ \leq 0, & j \in I_{\text{non-binding}}, \end{cases} \tag{125}$$

where $H_{\theta\theta}$ is the Hessian (or generalized curvature) at $(\bar{\varepsilon}, \bar{\theta}, \bar{\lambda})$, $b = \nabla_{\theta\varepsilon} L(\bar{\varepsilon}, \bar{\theta}, \bar{\lambda})\, \Delta\bar{\varepsilon}$, and each constraint has a cached linear direction

$$a_j \;:=\; \frac{1}{N_j} \sum_{i=1}^{N_j} \nabla_\theta \ell_j \left( z_i^{(j)}, \bar{\theta} \right).$$

Let $N_c = |I_{\text{binding}} \cup I_{\text{non-binding}}|$ be the total number of (potentially) active constraints. Collect all constraint directions into $A \in \mathbb{R}^{N_c \times d}$, where the $j$-th row is $a_j^\top$, and let $\eta \in \mathbb{R}^{N_c}$ contain the constant terms $\eta_j$.

**Slack and dual variables.**   Introducing slack variables $\zeta \in \mathbb{R}^{N_c}$ and dual variables $\xi \in \mathbb{R}^{N_c}$, the inequality constraints $A\omega + \eta \leq 0$ become the primal feasibility condition

$$A\omega + \zeta + \eta = 0, \qquad \zeta > 0, \qquad \xi > 0.$$

**KKT residuals.**   Under a logarithmic barrier with parameter $\rho > 0$, the primal–dual KKT conditions are

$$r_{\text{dual}} = H_{\theta\theta}\omega + b + A^\top \xi,$$
$$r_{\text{pri}} = A\omega + \zeta + \eta,$$
$$r_{\text{cent}} = \Gamma\Lambda\mathbf{1} - \rho\,\mathbf{1},$$

where $\Gamma = \text{diag}(\zeta)$ and $\Lambda = \text{diag}(\xi)$.

**Newton system.**   A primal–dual interior-point iteration computes a Newton step $(\Delta\omega, \Delta\xi, \Delta\zeta)$ by solving the linearized KKT system

$$\begin{bmatrix} H_{\theta\theta} & A^\top & 0 \\ A & 0 & I \\ 0 & \Gamma & \Lambda \end{bmatrix} \begin{bmatrix} \Delta\omega \\ \Delta\xi \\ \Delta\zeta \end{bmatrix} = - \begin{bmatrix} r_{\text{dual}} \\ r_{\text{pri}} \\ r_{\text{cent}} \end{bmatrix}. \tag{126}$$

The full KKT matrix has dimension $(d + 2N_c) \times (d + 2N_c)$, but the slack block can be explicitly eliminated because both $\Gamma$ and $\Lambda$ are diagonal. Consequently, the effective Newton system reduces to a $(d + N_c)$-dimensional saddle-point matrix in $(\omega, \xi)$.

**Computational complexity.**   We do not form the KKT matrix explicitly. Instead, each Newton step is solved by a matrix-free Krylov method (e.g., MINRES or CG applied to the Schur complement), which only requires evaluating KKT–matrix–vector products.

Eliminating $\Delta\zeta$ from the KKT system yields the reduced equation

$$A\,\Delta\omega + \Delta\zeta = -r_{\text{pri}}, \qquad \Gamma\Delta\xi + \Lambda\Delta\zeta = -r_{\text{cent}},$$

which allows expressing $\Delta\zeta$ and $\Delta\xi$ as affine functions of $\Delta\omega$. Substituting these expressions into the first KKT block gives the symmetric positive-definite Schur complement

$$M_{sc} = H_{\theta\theta} + A^\top \Gamma^{-1}\Lambda A, \qquad b_\omega = -r_{\text{dual}} - A^\top \Gamma^{-1}\big(-r_{\text{cent}} + \Lambda r_{\text{pri}}\big).$$

and the Newton direction for the primal variable is obtained by solving $M_{sc}\Delta\omega = b_\omega$ with CG. Once $\Delta\omega$ is computed, the dual and slack directions follow from the back-substitution rules

$$\Delta\xi = \Gamma^{-1}(\Lambda A\Delta\omega - r_{\text{cent}} + \Lambda r_{\text{pri}}), \qquad \Delta\zeta = -r_{\text{pri}} - A\Delta\omega.$$

All required operations are matrix–vector products involving $H_{\theta\theta}$ (via one HVP), $A$, $A^\top$, and the diagonal matrices $\Gamma$ and $\Lambda$.

Each Krylov iteration requires evaluating the KKT–matrix action on a vector $(\Delta\omega, \Delta\xi, \Delta\zeta)$, which consists of:

- computing the Hessian–vector product $H_{\theta\theta}\Delta\omega$, with cost $\mathcal{O}(d)$ via double backpropagation;
- applying $A\Delta\omega$ and $A^\top\Delta\xi$, each costing $\mathcal{O}(N_c * d)$;
- diagonal operations $\Gamma\Delta\xi$ and $\Lambda\Delta\zeta$, with cost $\mathcal{O}(N_c)$.

Since typically $N_c \ll d$, the dominant cost per Krylov iteration is $\mathcal{O}(d)$.

Assume we take $K_{\text{pd}}$ iterations for the primal–dual interior-point method. The total computational cost is

$$\mathcal{O}\big(K_{\text{pd}}d\big),$$

which is linear in the model dimension $d$ and independent of the number of training samples.

# G  MACHINE UNLEARNING OF PHYSICAL INFORMED NEURAL NETWORK (PINN)

We further evaluate DIF on a machine unlearning task for a Physics-Informed Neural Network (PINN) model, following the formulation in (Shi et al., 2021). The PINN is trained on the NGSIM vehicle-trajectory dataset (Coifman & Li, 2017) to estimate traffic velocity from observed trajectories, with a PDE constraint to enforce that the solution respects the traffic flow theory.

Let $u_\theta(x, t)$ be the neural-network approximation of the velocity field. The PINN training problem can be written as [3]

$$\min_\theta \quad \frac{1}{N}\sum_{i=1}^N \big(u_\theta(x_i, t_i) - v_i\big)^2$$

subject to the PDE constraint (traffic-flow model)

$$\frac{\partial u_\theta}{\partial t}(x, t) + u_\theta(x, t)\frac{\partial u_\theta}{\partial x}(x, t) = 0, \quad \forall (x, t) \in \Omega,$$

where $(x_i, t_i, v_i)$ are the observed NGSIM trajectory data and $\Omega$ is the spatio-temporal domain.

Figure 7 illustrates (i) the raw vehicle trajectories, (ii) the removal of a subset of the trajectory data to simulate unlearning, and (iii) the corresponding velocity field estimated by the PINN.

After removing 10% training data, we use DIF to estimate the change in the solution and obtain the updated model parameters. Table 2 presents the performance of three PINN models–Original, Retrained, and Unlearned–on the residual dataset after data removal. The Retrained Model achieves the best performance across all metrics, with the lowest data loss (MAE: 2.495 ), physics loss (MAE:

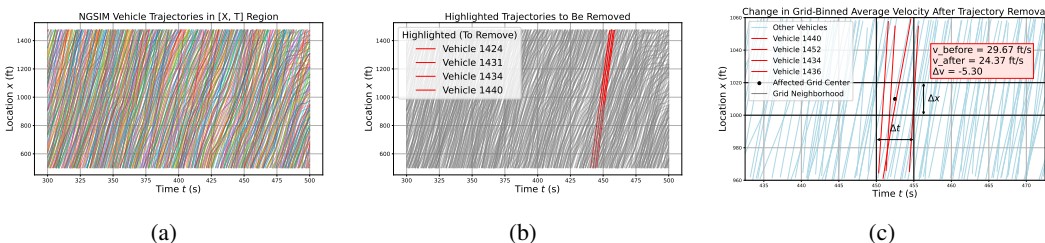

Figure 7: Illustration of trajectory removal and its impact on observed velocity. (a) All trajectories. (b) Subset of removed trajectories. (c) Change in average velocity in affected spatiotemporal bins due to trajectory removal.

Table 2: Performance comparison of different PINN models on the residual dataset after removal.

| Model | MAE (Data Loss) | MAE (Physics Loss) | Relative $L_2$ Error (%) | Training Time (s) |
|---|---|---|---|---|
| Original Model | 4.156 | $47.1 \times 10^{-2}$ | 21.65 | 327.06 |
| Retrained Model | 2.495 | $1.12 \times 10^{-2}$ | 14.24 | 331.50 |
| Unlearned Model | 2.832 | $0.37 \times 10^{-2}$ | 16.68 | 91.77 |

$1.12 \times 10^{-2}$ ), and relative $L_2$ error ( $14.24\%$ ). The Unlearned Model shows slightly higher errors than the Retrained Model, but remains significantly better than the Original Model.

# H    USE OF LARGE LANGUAGE MODELS (LLMS)

Large language models (e.g., ChatGPT, OpenAI) were used only to support writing, specifically for grammar correction and stylistic polishing. They were not involved in formulating research ideas, designing methods, analyzing results, or drawing conclusions. The authors remain fully responsible for all content of the paper.

---

[3]For simplicity of presentation in Appendix G, we reuse the notation $(x, t, v)$ to denote the spatial coordinate, timestamp, and observed velocity in the NGSIM data. These symbols are independent of their usage in the main text.

