# OpenReview forum: "Directional Influence Function: Estimating Training Data Influence in Constrained Learning"
_ICLR.cc/2026/Conference — Submitted to ICLR 2026_

### Official Review · Reviewer_wcqU · 2025-10-19

**Soundness:** 2
**Presentation:** 2
**Contribution:** 3
**Rating:** 6
**Confidence:** 3

**Summary:**

This paper extends the Influence Function (IF) framework to the setting of constrained learning. The authors first demonstrate, through a toy example, that the classical IF can fail under such constraints. To address this limitation, they formulate the problem using variational inequality (VI) theory to analyze how perturbations in the training data affect the solution, and subsequently derive the proposed Directional Influence Function (DIF) by solving a quadratic programming (QP) problem. The proposed method is evaluated on constrained linear regression and the MNIST dataset.

**Strengths:**

1. This paper investigates Influence Functions under the setting of constrained learning, which introduces a novel and interesting aspect to the study of IF.
2. The motivation of this work is strong, and the toy example effectively illustrates the limitations of existing Influence Functions under the constrained learning setting.
3. This work introduces variational inequality (VI) to formulate constrained learning and analyzes how perturbations in the training data affect this VI, thereby providing a novel analytical perspective on Influence Functions.

**Weaknesses:**

1. The experimental evaluation is rather limited. Specifically, the authors only conduct experiments on constrained linear regression and the MNIST dataset using a simple neural network with several convolutional layers. To more convincingly demonstrate the effectiveness of the proposed method, additional experiments on downstream tasks, such as noisy label identification and sample selection, are necessary.

**Questions:**

1. Regarding the experiment on the constrained CNN, could the authors provide the results obtained by the classical IF for a clearer comparison between the proposed DIF and IF?
2. Since the current applications of IF are primarily in deep neural networks with large-scale parameters, how computationally expensive is the proposed QP problem (Equation 18)? Could the authors provide some discussion on how this QP problem can be efficiently solved in practice?

---

> ### Author Response · Authors · 2025-11-20
> **Response to Reviewer wcqU (Part 1/2)**
>
> >W1. The experimental evaluation is rather limited. Specifically, the authors only conduct experiments on constrained linear regression and the MNIST dataset using a simple neural network with several convolutional layers. To more convincingly demonstrate the effectiveness of the proposed method, additional experiments on downstream tasks, such as noisy label identification and sample selection, are necessary.
>
> A. Thank you for the suggestion. We agree that broader downstream evaluation is useful for demonstrating the effectiveness of DIF. We have added two downstream applications that directly address the reviewer’s concerns.
>
> **(1) Sample selection / harmful-point identification.  **
>
> We evaluate DIF on identifying harmful training samples by comparing it with
> TracIn [1] and TRAK [2] under counterfactual retraining. For each estimator, we rank
> training samples, remove the most harmful 2% and 10%, and retrain the model.
> DIF consistently achieves the largest loss reduction on a fixed misclassified
> test sample, demonstrating that it accurately identifies harmful points in a
> constrained training setting. We have added the corresponding experimental details in Section 5.2.
>
> **Table 1. Loss reduction after removing 2% harmful points (mean ± std).**
>
> | Method | Absolute Loss Reduction |
> |--------|--------------------------|
> | DIF | 0.0143 ± 0.0011 |
> | TracIn | 0.0113 ± 0.0003 |
> | TRAK | 0.0030 ± 0.0002 |
> | Random | 0.0001 ± 0.0011 |
>
> **Table 2. Loss reduction after removing 10% harmful points (mean ± std).**
>
> | Method | Absolute Loss Reduction |
> |--------|--------------------------|
> | DIF | 0.0492 ± 0.0008 |
> | TracIn | 0.0452 ± 0.0001 |
> | TRAK | 0.0098 ± 0.0001 |
> | Random| 0.0010 ± 0.0033 |
>
> **(2) Machine unlearning on PINNs (physical informed Neural Networks).  **
>
> We applied it to a machine unlearning problem for Physics-Informed Neural Networks (PINNs) [3].
> For clarity, the PINN used in our unlearning experiment can be written as a constrained optimization problem. Let $u_\theta(x,t)$ denote the neural-network approximation of the velocity field. The training problem is
>
> $\min_\theta \; \frac{1}{N}\sum_{i=1}^N \big( u_\theta(x_i,t_i) - v_i \big)^2$
>
> subject to the PDE constraint (traffic-flow model)
> $$
> \frac{\partial u_\theta}{\partial t}(x, t)+u_\theta(x, t) \frac{\partial u_\theta}{\partial x}(x, t)=0, \quad \forall(x, t) \in \Omega,
> $$
> for all $(x,t)$ in the spatio-temporal domain. Here $(x_i,t_i,v_i)$ are the observed positions, timestamps, and velocities from the NGSIM trajectory data.
>
> In this experiment, the PINN is trained on the NGSIM dataset [4], a vehicle-trajectory dataset, to estimate traffic velocity. The model is a multilayer perceptron equipped with a PDE-based physical constraint, which ensures the predicted solution remains consistent with fundamental traffic-flow theory. After removing 10% of the training trajectories, we use DIF to estimate the parameter update and obtain the corresponding unlearned model. We then compare this DIF-estimated model with full retraining.
>
>
>
>
>
>
>
>
>
> **Table 3. Performance comparison of different PINN models on the residual dataset after data removal.**
>
> | Model            | MAE (Data Loss) | MAE (Physics Loss) | Relative L2 Error (%) |
> |------------------|------------------|----------------------|------------------------|
> | Original Model  | 4.156            | 4.71e-1              | 21.65                 |
> | Retrained Model  | 2.495            | 1.12e-2              | 14.24                 |
> | Unlearned Model | 2.832            | 3.70e-3              | 16.68                 |
>
> [1] Pruthi et al. *Estimating training data influence by tracing gradient descent*. NeurIPS 2020.
> [2] Park et al. *TRAK: Attributing model behavior at scale*. arXiv:2303.14186.
>
> [3] Shi, Rongye, Zhaobin Mo, and Xuan Di. "Physics-informed deep learning for traffic state estimation: A hybrid paradigm informed by second-order traffic models." Proceedings of the AAAI Conference on Artificial Intelligence. Vol. 35. No. 1. 2021.
>
> [4] FHWA. Next Generation Simulation (NGSIM) Vehicle Trajectory Data. U.S. Department of Transportation, Federal Highway Administration, 2007.

---

> > ### Author Response · Authors · 2025-11-20
> > **Response to Reviewer wcqU (Part 2/2)**
> >
> > >Q1. Regarding the experiment on the constrained CNN, could the authors provide the results obtained by the classical IF for a clearer comparison between the proposed DIF and IF?
> >
> > A1. Thank you for the question. We have added the results of the classical penalty-based influence function (Penalty IF) on the constrained CNN. Specifically, we applied IF to the penalized objective and tuned the penalty coefficient. Under this setting, the predicted loss change and the actual loss change achieve a correlation of approximately 0.62, which is substantially lower than the 0.90 correlation achieved by DIF.
> >
> > Moreover, computing the IF term $H^{-1}\nabla_{\theta} L(x_{\mathrm{train}}, \bar{\theta})$ with conjugate gradient (CG) proved difficult in practice. The CG iterations often failed to converge unless we repeatedly adjusted the damping parameter and the penalty coefficient, indicating high sensitivity to hyperparameter choices in the constrained CNN setting.
> > These results and the comparison with DIF have been added to Section 5.2 of the revised manuscript
> >
> > >Q2. Since the current applications of IF are primarily in deep neural networks with large-scale parameters, how computationally expensive is the proposed QP problem (Equation 18)?
> > Could the authors provide some discussion on how this QP problem can be efficiently solved in practice?
> >
> > A2. Thank you for the suggestion. Specifically, DIF computes the update by solving a quadratic program in the variable $ \omega $, which we optimize using a primal–dual method.  The solution $ \omega^\ast $ of this quadratic program converges to the desired parameter update $ \Delta \theta $. Let $ \bar{\theta} $ denote the original model parameters before data perturbation, with parameter dimension $ p $, and let $ N_c $ be the number of active constraints.
> >
> > At iteration $ t $, updating $\omega$ requires to compute the product between the Hessian at $ \bar{\theta} $ and the current vector $ \omega^{(t)} $, i.e., $\nabla_{\theta \theta}^2 L(\bar{\varepsilon}, \bar{\theta}, \bar{\lambda}) \omega$. Using an HVP, this operation takes $O(p)$ time.
> >
> > We also need to compute i) the products between the gradient of the objective at $ \bar{\theta}$ with $omega$ and dual variables, e.g., $\nabla \theta L\left(x_{\text{train}}, \bar{\theta}\right) \cdot \omega$ and (ii) products between the Jacobian of the constraints at $ \bar{\theta} $, with $ \omega^{(t)} $ and the dual variables, e.g., $\nabla_\theta \ell_j \left(x_{\text{train}}, \bar{\theta}\right) \cdot \omega$. The gradients at $ \bar{\theta} $ can be precomputed and reused. This step costs $O(N_c p)$. Since $ N_c \ll p $, the per-iteration complexity remains $O(p)$.
> >
> > Thus, each primal–dual iteration costs $O(p)$. If the primal–dual solver performs $K$ iterations, the overall DIF takes $O(Kp)$ time.
> >
> > As a comparison, standard influence functions (IF) require only $O(p)$ time for a single training sample. For large scale models, DIF scales linearly with the number of parameters.
> >
> > We have added a detailed derivation in Appendix F, where we explicitly present the computations in each iteration and analyze their corresponding costs.

---

### Official Review · Reviewer_4m1P · 2025-10-26

**Soundness:** 4
**Presentation:** 2
**Contribution:** 3
**Rating:** 6
**Confidence:** 4

**Summary:**

The paper introduces the Directional Influence Function (DIF), a generalization of the classical influence function for constrained learning problems. The authors argue that standard influence functions fail in this setting because (i) they ignore feasibility preservation and (ii) the derivative of the optimal solution with respect to data perturbations becomes ill-defined near solutions with active inequality constraints. DIF resolves these issues and can be computed by solving a quadratic program for each datapoint. It reduces to the classical influence function when no constraints are active and satisfies first-order correctness, accurately predicting the effect of small data perturbations. Experiments on norm-constrained linear regression and fairness-constrained CNNs show that DIF closely matches the true influence observed through leave-one-out retraining.

**Strengths:**

**Originality**. IF-style analysis for constrained learning is novel in ML (though precedents exist in statistics). The novel variational-inequality treatment handles non-smoothness from active constraints and yields a practical recipe: a single QP per point that, in principle, implicitly handles active-set selection after perturbation.

**Quality**. The theoretical results seem sound. The empirical evidence is positive: DIF nearly matches leave-one-out on constrained linear regression and shows strong correlation on the fairness-constrained CNN.

**Clarity**. The paper is decently easy to follow, and the toy constrained-regression example (Fig. 1) effectively illustrates why classical IF can fail.

**Significance**. The work addresses a timely need for interpretability in constrained learning—a field of growing importance for ensuring fairness, safety, and robustness in ML systems—and provides a practical method for computing directional influence functions.

**Weaknesses:**

Additional work is needed to clearly establish the paper’s limitations, substantiate its novelty, and ensure the overall scientific rigor of the submission. I am willing to increase my score if the following points are addressed.
### Major complaints

1. **Implicit Active-set Stability Assumption**. The paper argues that classical IF fails in constrained learning because removing a data point can change the active constraint set and cause non-differentiable jumps in the solution. However, the DIF derivation still fixes the current active set (binding / non-binding / inactive), applies different linearized rules to each group, and assumes that this partition continues to govern the local response under perturbation. This amounts to an implicit active-set stability assumption. The paper should state this explicitly and clarify when it may fail (e.g., when an inactive constraint is about to become active), rather than presenting the guarantees as generally applicable.


2. **Computational cost**. The paper should analyze and compare the computational costs of IF and DIF, and report these measurements in the experimental section.


3. **Baselines on CNNs**. The authors should include baselines such as the classical IF and penalty-IF on their CNN task experiments. As it stands, DIF’s performance looks reasonable, but without these baselines it’s impossible to quantify DIF’s relative improvement over IF/penalty-IF.


4. **Penalty-IF tuning**. While penalty-IF is not a principled approach to constrained influence functions, the authors should ensure that its experimental setup is rigorous. The choice of the penalty coefficient for penalty-IF in Figure 3 is not reported, nor is the procedure for selecting it. This omission raises the risk that penalty-IF was under-tuned, potentially overstating DIF’s practical improvement.

5. **Related Works**. The paper fails to acknowledge any prior research on influence functions for constrained estimators. However, some works along these lines exist [1, 2]. While the paper’s treatment via a Variational Inequality formulation is novel, these works should be discussed to contextualize the contribution. Moreover, the related works section (currently in Appendix A) should be moved to the main body to properly situate the paper within this existing literature and clarify what is genuinely new.
    - [1] Abhik Ghosh. Influence function analysis of the restricted minimum divergence estimators: A general form. Electronic Journal of Statistics, 2015.
    - [2] Klaus L.P. Vasconcellos, L.M. Zea Fernandez. Influence analysis with homogeneous linear restrictions. Computational Statistics & Data Analysis, 2009.

6. **Scope of the theoretical guarantees in deep learning**. Important results depend on LICQ, SOSC and the stability of the active-set at the KKT point (see point 1). These regularity conditions are not automatically satisfied in deep networks, which often exhibit flat directions and active constraints right at activation boundaries. The paper should explicitly acknowledge this limitation and clarify that the theorems may not strictly apply to generic deep-learning models.

### Minor complaints
7. **Handling of Equality Constraints**. The paper briefly notes that equality constraints can be converted into inequalities (e.g., ($h(x)=0$) becomes ($h(x) \le 0$) and ($-h(x) \le 0$)). While this is correct, it doubles the number of constraints and can complicate regularity assumptions such as LICQ, which are crucial for several of the paper’s theoretical results. A more direct treatment of equality constraints within the VI framework would have been cleaner.

8. **Clarity and Structure**. A few suggestions could enhance readability. The core theory in Section 4.2—particularly the “Auxiliary VI” (Eq. 15) and its proof—relies on concepts such as the “Critical Cone” (Definition 15), which are defined only in Appendix B. These definitions should be moved into the main text for clarity. In addition, the paper transitions abruptly from defining DIF as a directional derivative (Section 3) to deriving it via Variational Inequalities (Section 4) without explaining the conceptual connection or motivating why the VI framework is appropriate. Finally, Section 3 is unnecessarily long for its content, dedicating a full page to two simple definitions and a corollary; it should be condensed.

9. **Presentation and Notation**. The paper would benefit from some additional polish. For instance:
    - In Section 2.1, the text refers to an “$\ell_1$-boundary,” while the problem itself is defined with an $\ell_2$ norm.


   - Equation (76) uses $\Delta \lambda$ (missing hat) and Equation (85) uses ($\hat{\lambda}$) (missing $\Delta$).


    - Theorem 10 uses ($w^{\*}$) instead of the ($\omega^{\*}$) notation used in the QP.


    - The proof of Proposition 12 incorrectly cites “Proposition 10” instead of Theorem 10.

**Questions:**

1. Why were the standard IF and penalty-IF baselines not included in the CNN experiment? Without them, it is difficult to assess DIF’s relative improvement.


2. In the CNN experiment, was the QP used to compute the DIF solved exactly? Given that the Hessian term is fairly high-dimensional, did you use any approximation to reduce computational cost? Which solver did you use?


3. What were the values of the penalty coefficients used for the penalty-IF baseline in Figure 3, and how were they chosen (e.g., grid search, manually)?

---

> ### Author Response · Authors · 2025-11-21
> **Response to Reviewer 4m1P (Part 1/3)**
>
> >W1. Implicit Active-set Stability Assumption.
>
> A1. We thank the reviewer for this observation. Importantly, our method does not assume a fully rigid active-set stability. Instead, we use a weaker and standard form of stability derived from complementarity. In particular, at the reference solution $\bar{\theta}$ in the auxiliary problem (18):
>
> •if $g_j(\bar{\theta}) = 0$ and $\bar{\lambda}_j > 0$ (the binding set), we require the constraint to remain an equality. The multiplier is strictly positive and can adjust slightly to absorb the perturbation, keeping the constraint exactly on the boundary without violating the KKT conditions.
>
> •if $g_j(\bar{\theta}) = 0$ but $\bar{\lambda}_j = 0$ (the non-binding set), we only require $g_j(\theta(\varepsilon)) \le 0$ and do not force the constraint to remain at zero. Once the multiplier is zero, allowing the constraint to become strictly inactive ($g_j(\theta(\varepsilon)) < 0$) does not violate complementarity.
>
> •inactive constraints place no restrictions.
>
> **Thus, constraints that are active but non-binding are allowed to move into the slack region under infinitesimal perturbations.**
>
> Moreover, the assumptions of Theorem 8 ensure local Lipschitz continuity of the solution mapping with respect to data perturbations, which implies that both the primal variables and the dual variables vary continuously with the data. As a result, no jump discontinuity arises in either the constraint values or the multipliers under small perturbations. If the two assumptions do not hold, the binding or non-binding sets may change even when the solution moves infinitesimally away from $\bar{\theta}$. In such cases, the proposed method may not apply. We have added these clarifications to Section 4.2 after Theorem 8.
>
>
>
> >W2. Computational cost. The paper should analyze and compare the computational costs of IF and DIF, and report these measurements in the experimental section.
>
> A2. Thank you for the suggestion. Specifically, the computational cost of DIF can be analyzed as follows.
>
> DIF computes the update by solving a quadratic program in the variable $ \omega $, which we optimize using a primal–dual method.  The solution $ \omega^\ast $ of this quadratic program converges to the desired parameter update $ \Delta \theta $. Let $ \bar{\theta} $ denote the original model parameters before data perturbation, with parameter dimension $ p $, and let $ N_c $ be the number of active constraints.
>
> At iteration $ t $, updating $\omega$ requires to compute the product between the Hessian at $ \bar{\theta} $ and the current vector $ \omega^{(t)} $, i.e., $\nabla_{\theta \theta}^2 L(\bar{\varepsilon}, \bar{\theta}, \bar{\lambda}) \omega$. Using an HVP, this operation takes $O(p)$ time.
>
> We also need to compute i) the products between the gradient of the objective at $ \bar{\theta}$ with $omega$ and dual variables, e.g., $\nabla \theta L\left(x_{\text{train}}, \bar{\theta}\right) \cdot \omega$ and (ii) products between the Jacobian of the constraints at $ \bar{\theta} $, with $ \omega^{(t)} $ and the dual variables, e.g., $\nabla_\theta \ell_j \left(x_{\text{train}}, \bar{\theta}\right) \cdot \omega$. The gradients at $ \bar{\theta} $ can be precomputed and reused. This step costs $O(N_c p)$. Since $ N_c \ll p $, the per-iteration complexity remains $O(p)$. Thus, each primal–dual iteration costs $O(p)$. If the primal–dual solver performs $K$ iterations, the overall DIF takes $O(Kp)$ time.
>
> As a comparison, standard influence functions (IF) require only $O(p)$ time for a single training sample. For large scale models, DIF relies only on Hessian–vector products and Jacobian–vector products, both of which are implemented efficiently in modern autograd systems and scale linearly with the number of parameters.
> We have added a detailed derivation in Appendix F, where we explicitly present the computations in each iteration and analyze their corresponding costs.
> > W3. Baselines on CNNs. The authors should include baselines such as the classical IF and penalty-IF on their CNN task experiments.
>
> A3. Thank you for the question. We have added the results of the classical penalty-based influence function (Penalty IF) on the constrained CNN. Specifically, we applied IF to the penalized objective and tuned the penalty coefficient. Under this setting, the predicted loss change and the actual loss change achieve a correlation of approximately 0.62, which is substantially lower than the 0.90 correlation achieved by DIF.
>
> Moreover, computing the IF term $H^{-1}\nabla_{\theta} L(x_{\mathrm{train}}, \bar{\theta})$ with conjugate gradient (CG) proved difficult in practice. The CG iterations often failed to converge unless we repeatedly adjusted the damping parameter and the penalty coefficient, indicating high sensitivity to hyperparameter choices in the constrained CNN setting.
> These results and the comparison with DIF have been added to Section 5.2 of the revised manuscript.

---

> > ### Author Response · Authors · 2025-11-21
> > **Response to Reviewer 4m1P (Part 2/3)**
> >
> > >W4. Penalty-IF tuning. While penalty-IF is not a principled approach to constrained influence functions, the authors should ensure that its experimental setup is rigorous. The choice of the penalty coefficient for penalty-IF in Figure 3 is not reported, nor is the procedure for selecting it. This omission raises the risk that penalty-IF was under-tuned, potentially overstating DIF’s practical improvement.
> >
> > A4. Thank you for raising this concern. The penalty coefficient used in Figure 3 was in fact carefully tuned. We have added the full tuning procedure to Appendix D.
> >
> > Denote the solution before data perturbation as $ \bar{\theta} $. For the constrained linear regression problem in Figure 3, the inequality constraint is inactive at $ \bar{\theta} $. This implies that the corresponding penalty term does not affect the influence score. For the equality constraint, our grid search tuning procedure shows that increasing the penalty coefficient improves the accuracy of the predicted loss change, but the gains saturate once the coefficient exceeds roughly 10. Based on this trend, we selected a coefficient of 200. We have documented this selection process and the associated empirical observations in Appendix D to ensure full transparency.
> >
> > >W5. Related Works.
> >
> > A5.  Thank you for highlighting these relevant works. We have incorporated them into the revised related-work section, and we will move this section from the appendix to the main body in the camera-ready version.
> >
> > >W6. Scope of the theoretical guarantees in deep learning. Important results depend on LICQ, SOSC and the stability of the active-set at the KKT point (see point 1). These regularity conditions are not automatically satisfied in deep networks, which often exhibit flat directions and active constraints right at activation boundaries. The paper should explicitly acknowledge this limitation and clarify that the theorems may not strictly apply to generic deep-learning models.
> >
> > A6. We have revised the manuscript to explicitly acknowledge this limitation and to clarify these regularity conditions may not strictly apply to all neural networks, especially those with flat directions or activation-boundary effects.
> > Minor complaints
> >
> > >W7. Handling of Equality Constraints. The paper briefly notes that equality constraints can be converted into inequalities. While this is correct, it doubles the number of constraints and can complicate regularity assumptions such as LICQ, which are crucial for several of the paper’s theoretical results. A more direct treatment of equality constraints within the VI framework would have been cleaner.
> >
> > A7. Thank you for pointing this out. In the current version of the manuscript, we focused on inequality constraints because they are widely used in constrained learning, including invariance learning, robust learning, and fairness learning. Equality constraints can also be handled in our VI framework by naturally introducing additional dual variables, and doing so does not require any conceptual change to the method.
> >
> > >W8. Clarity and Structure.
> >
> > A8. Thank you for these helpful suggestions. We agree that moving the relevant definitions (e.g., the critical cone) into the main text and improving the transition between Sections 3 and 4 would further enhance readability. We will incorporate a full VI formulation that handles equality constraints directly in the extended version of the paper.
> >
> > >W9. Presentation and Notation.
> >
> > A9. Thank you for pointing out these presentation and notation issues. We have
> > corrected all of them in the revised version

---

> > > ### Author Response · Authors · 2025-11-21
> > > **Response to Reviewer 4m1P (Part 3/3)**
> > >
> > > >Q1. Why were the standard IF and penalty-IF baselines not included in the CNN experiment? Without them, it is difficult to assess DIF’s relative improvement.
> > >
> > > A1. Thank you for the question. This point is already addressed in our response to W3
> > >
> > > >Q2. In the CNN experiment, was the QP used to compute the DIF solved exactly? Given that the Hessian term is fairly high-dimensional, did you use any approximation to reduce computational cost? Which solver did you use?
> > >
> > > A2. Thank you for the question. In the CNN experiment, we do not solve the QP in closed form. Instead, we approximate its solution using a primal–dual method. At iteration $t$, the update requires computing $\nabla_{\theta\theta}^2 L(\bar{\theta})\,\omega^{(t)}$, which is obtained efficiently through a Hessian–vector product (HVP). The Hessian–vector product (HVP) takes $O(p)$ time, where $p$ denotes the dimension of the parameter vector.
> > >
> > > For each iteration, we also compute the Jacobian–vector products associated with the constraints. For each active constraint $\ell_j(\theta)$, the primal–dual update requires evaluating $\nabla_\theta \ell_j(\bar{\theta})^\top \omega^{(t)}$ and its dual-weighted counterpart. These are obtained via standard Jacobian–vector products，which autograd computes using a single reverse-mode pass. Thus, each primal–dual iteration costs $O(p)$. If the primal–dual solver performs $K$ iterations, the overall DIF takes $O(Kp)$ time.
> > > As a comparison, standard influence functions (IF) require $O(p)$ time for a single training sample.
> > >
> > > >Q3.	What were the values of the penalty coefficients used for the penalty-IF baseline in Figure 3, and how were they chosen (e.g., grid search, manually)?
> > >
> > > A. Thank you for the question. This point is already addressed in our response to W4

---

> ### Comment · Reviewer_4m1P · 2025-11-22
> **Response acknowledgement and score raised to 8**
>
> I thank the authors for their detailed responses and the substantial improvements made to the manuscript.
>
> Given that my complaints have been thoroughly addressed, I am raising my score from a 6 to an 8.

---

### Official Review · Reviewer_DRab · 2025-10-30

**Soundness:** 3
**Presentation:** 4
**Contribution:** 3
**Rating:** 8
**Confidence:** 3

**Summary:**

This paper introduces the DIF, which is a new framework for estimating how individual training samples influence model parameters in constrained learning settings, where constraints arise from fairness, safety, robustness, etc.. It shows that the classical influence function becomes unreliable because data perturbations can lead to infeasible or biased estimates. Authors formulate constrained learning as a variational inequality and derive DIF through directional sensitivity analysis, ensuring that estimated parameter updates remain within the feasible region. Experiments on constrained linear regression and fairness constrained CNNs demonstrate that DIF closely matches LOO results, outperforming both IF and penalty based IF estimators in accuracy and feasibility.

**Strengths:**

1. The paper introduces the DIF, a principled and general framework for analyzing sample influence in constrained learning, which has not been adequately handled by classical influence functions. DIF explicitly accounts for active constraints through VI framework, ensuring that estimated parameter perturbations remain feasible, which a significant advancement over prior IF methods that often produce infeasible updates. I think it is a novel theoretical contribution. Authors also provides a clear theoretical foundation, making the approach both interpretable and well grounded.
2. DIF offers a unified framework that can extend to many practical settings i.e., fairness, safety and robust learning. The paper also points to promising applications such as data poisoning detection, unlearning, and online constraint adaptation, indicating strong future relevance.

**Weaknesses:**

1. Although DIF reduces to solving a quadratic program, solving a QP for each influence data can still be computationally expensive compared to classical IF approximations, particularly for deep networks with large amount of parameters.
2. The baselines are primarily classical IF and penalty-based IF; some recent scalable or robust IF variants (e.g., TracIn, Hessian-free methods) are not included, limiting the completeness of the empirical comparison.

**Questions:**

How does DIF scale when applied to large deep neural networks with large number of parameters and complex constraint structures?

---

> ### Author Response · Authors · 2025-11-20
> **Response to Reviewer  DRab (Part 1/2)**
>
> >W1. Although DIF reduces to solving a quadratic program, solving a QP for each influence data can still be computationally expensive compared to classical IF approximations, particularly for deep networks with large amount of parameters.
>
> A1. Thank you for the suggestion. We have added a detailed derivation in Appendix F, where we explicitly present the computations in each iteration and analyze their corresponding costs. Specifically, the computational cost of DIF can be analyzed as follows.
>
> DIF computes the update by solving a quadratic program in the variable $ \omega $, which we optimize using a primal–dual method.  The solution $ \omega^\ast $ of this quadratic program converges to the desired parameter update $ \Delta \theta $. Let $ \bar{\theta} $ denote the original model parameters before data perturbation, with parameter dimension $ p $, and let $ N_c $ be the number of active constraints.
>
> At iteration $ t $, updating $\omega$ requires to compute the product between the Hessian at $ \bar{\theta} $ and the current vector $ \omega^{(t)} $, i.e., $\nabla_{\theta \theta}^2 L(\bar{\varepsilon}, \bar{\theta}, \bar{\lambda}) \omega$. Using an Hessian–vector products (HVP), this operation takes $O(p)$ time.
>
> We also need to compute i) the products between the gradient of the objective at $ \bar{\theta}$ with $omega$ and dual variables, e.g., $\nabla \theta L\left(x_{\text{train}}, \bar{\theta}\right) \cdot \omega$ and (ii) products between the Jacobian of the constraints at $ \bar{\theta} $, with $ \omega^{(t)} $ and the dual variables, e.g., $\nabla_\theta \ell_j \left(x_{\text{train}}, \bar{\theta}\right) \cdot \omega$. The gradients at $ \bar{\theta} $ can be precomputed and reused. This step costs $O(N_c p)$. Since $ N_c \ll p $, the per-iteration complexity remains $O(p)$.
>
> Thus, each primal–dual iteration costs $O(p)$. If the primal–dual solver performs $K$ iterations, the overall DIF takes $O(Kp)$ time.
> As a comparison, standard influence functions (IF) require only $O(p)$ time for a single training sample. For large scale models, DIF relies only on Hessian–vector products and Jacobian–vector products, both of which are implemented efficiently in modern autograd systems and scale linearly with the number of parameters.
>
> >W2. The baselines are primarily classical IF and penalty-based IF; some recent scalable or robust IF variants (e.g., TracIn, Hessian-free methods) are not included, limiting the completeness of the empirical comparison.
>
>
> A2. We evaluate counterfactual retraining using three influence estimators: DIF, TracIn [1], and TRAK [2]. For each method, we compute per-sample influence scores on the training set, remove the most harmful 2% and 10% samples, and then retrain the model. We then compare the resulting loss changes on a fixed misclassified test point. The constrained CNN is trained using a primal–dual method.
> For TracIn, we develop a constrained variant following prior work. During primal–dual updates, we record the gradients of the Lagrangian loss with respect to the model parameters. The TracIn score for a training–test pair is computed as $ \mathrm{TracIn}(x_{\mathrm{train}}, x_{\mathrm{test}}) = \sum_{i=1}^{k} \delta_i \, \nabla \ell_{\mathrm{lag}}(\theta_{t_i}, x_{\mathrm{train}}) \cdot\nabla \ell_{\mathrm{lag}}(\theta_{t_i}, x_{\mathrm{test}}), $, where $ \delta_i $ is the step size between checkpoints $ i{-}1 $ and $ i $, and $ \{\theta_{t_1}, \theta_{t_2}, \ldots, \theta_{t_k}\} $ are the parameter checkpoints recorded along the primal–dual trajectory. For TRAK, we apply the method to the penalized objective, where the constraints are incorporated into the loss using a squared hinge penalty.
>
> Both DIF and TracIn achieve substantial loss reductions, with DIF slightly outperforming TracIn, indicating that they can effectively identify the most harmful samples under our constrained training setup. In contrast, TRAK provides little improvement because its random-projection approximation fails to preserve the constraint-adjusted gradient directions, leading to ineffective influence estimates. We have added a detailed discussion of these results in Section 5.2 of the revised manuscript.
>
>
> **Table 1. Loss reduction after removing 2% harmful points (mean ± std).**
> | Method | Absolute Loss Reduction |
> |--------|--------------------------|
> | DIF | 0.0143 ± 0.0011 |
> | TracIn | 0.0113 ± 0.0003 |
> | TRAK | 0.0030 ± 0.0002 |
> | Random | 0.0001 ± 0.0011 |
>
> **Table 2. Loss reduction after removing 10% harmful points (mean ± std).**
>
> | Method | Absolute Loss Reduction |
> |--------|--------------------------|
> | DIF| 0.0492 ± 0.0008 |
> | TracIn | 0.0452 ± 0.0001 |
> | TRAK | 0.0098 ± 0.0001 |
> | Random | 0.0010 ± 0.0033 |
>
> [1] Pruthi et al. *Estimating training data influence by tracing gradient descent*. NeurIPS 2020.
>
> [2] Park et al. *TRAK: Attributing model behavior at scale*. arXiv:2303.14186.

---

> ### Author Response · Authors · 2025-11-20
> **Response to Reviewer DRab (Part 2/2)**
>
> >Q. How does DIF scale when applied to large deep neural networks with large number of parameters and complex constraint structures?
>
> A. Thank you for the follow-up question. Beyond the per-iteration complexity analysis, DIF is designed to scale to large deep networks because it does not require forming or storing any second-order tensors. All second-order information enters only through Hessian–vector products, which can be computed using standard autograd in linear time and constant memory overhead. This makes the method applicable to deep neural network models with large number of parameters without additional memory constraints.
>
> Regarding complex constraint structures, the QP only requires Jacobian–vector products with the active constraints. These Jacobian terms are computed using the same autograd mechanisms as the HVP and therefore scale linearly with the number of parameters, independent of the explicit size of the Jacobian. In practice, the number of active constraints is typically very small, so the additional overhead remains modest.

---

### Official Review · Reviewer_GE37 · 2025-11-04

**Soundness:** 3
**Presentation:** 3
**Contribution:** 2
**Rating:** 4
**Confidence:** 3

**Summary:**

This paper introduces the Directional Influence Function (DIF), a novel method for estimating the influence of training data in constrained learning settings. DIF addresses the limitations of classical Influence Functions (IF), which fail to account for constraints and feasibility requirements in constrained learning problems. By reformulating the optimality conditions as a Variational Inequality (VI) and leveraging sensitivity analysis, DIF provides a feasibility-preserving and efficient approach to estimate the impact of data perturbations. The authors validate DIF on constrained linear regression and fairness-constrained CNNs, demonstrating its accuracy and reliability compared to classical IF and penalty-based IF methods.

**Strengths:**

- The paper introduces DIF, a significant advancement over classical IF, specifically tailored for constrained learning problems. This is a valuable contribution to the field of machine learning interpretability and robustness.

- DIF is validated on two distinct tasks—constrained linear regression and fairness-constrained CNNs. The results show that DIF closely aligns with ground-truth retraining, outperforming classical IF and penalty-based IF in terms of accuracy and feasibility.  However, I have some comments on the models in the Weaknesses.

-  The paper demonstrates that DIF can be efficiently computed using Quadratic Programming (QP), making it practical for real-world applications.

**Weaknesses:**

- While the paper validates DIF on constrained linear regression and fairness-constrained CNNs, the experimental scope could be expanded to include other constrained learning scenarios, such as reinforcement learning with constraints.

- The paper primarily compares DIF with classical IF and penalty-based IF. It would be beneficial to include comparisons with other state-of-the-art methods for constrained learning or data attribution (e.g., TRAK).

- While the theoretical and empirical results are strong, the paper could provide more discussion on the practical implications of DIF, such as computational scalability for large-scale deep learning models.

- The paper briefly mentions the challenges of non-convex optimization in CNNs but does not delve deeply into how DIF handles non-convexity. A more detailed analysis would strengthen the paper.

**Questions:**

This paper makes a strong contribution to the field of constrained learning by introducing DIF, a theoretically sound and empirically validated method for data attribution. The work is rigorous and addresses a critical gap in existing influence estimation methods. However, the experimental scope and practical implications are weak at this point, therefore I am leaning towards a weak reject.

---

> ### Author Response · Authors · 2025-11-21
> **Response to Reviewer GE37 (Part 1/2)**
>
> >W1. While the paper validates DIF on constrained linear regression and fairness-constrained CNNs, the experimental scope could be expanded to include other constrained learning scenarios, such as reinforcement learning with constraints.
>
> A1. Thank you for this insightful suggestion. To broaden the applicability of DIF, we applied it to a machine unlearning problem for Physics-Informed Neural Networks (PINNs) [1].
> For clarity, the PINN used in our unlearning experiment can be written as a constrained optimization problem. Let $u_\theta(x,t)$ denote the neural-network approximation of the velocity field. The training problem is
> $\min_\theta \; \frac{1}{N}\sum_{i=1}^N \big( u_\theta(x_i,t_i) - v_i \big)^2$
> subject to the PDE constraint (traffic-flow model)
> $$
> \frac{\partial u_\theta}{\partial t}(x, t)+u_\theta(x, t) \frac{\partial u_\theta}{\partial x}(x, t)=0, \quad \forall(x, t) \in \Omega,
> $$
> for all $(x,t)$ in the spatio-temporal domain. Here $(x_i,t_i,v_i)$ are the observed positions, timestamps, and velocities from the NGSIM trajectory data.
> In this experiment, the PINN is trained on the NGSIM dataset [2], a vehicle-trajectory dataset, to estimate traffic velocity. The model is a multilayer perceptron equipped with a PDE-based physical constraint, which ensures the predicted solution remains consistent with fundamental traffic-flow theory. After removing 10% of the training trajectories, we use DIF to estimate the parameter update and obtain the corresponding unlearned model. We then compare this DIF-estimated model with full retraining.
>
> **Table 1. Performance comparison of different PINN models on the residual dataset after data removal.**
>
> | Model | MAE (Data Loss) | MAE (Physics Loss) | Relative L2 Error (%) |
> |------------------|------------------|----------------------|------------------------|
> | Original Model  | 4.156  | 4.71e-1  | 21.65   |
> | Retrained Model | 2.495  | 1.12e-2  | 14.24  |
> | Unlearned Model | 2.832  | 3.70e-3   | 16.68 |
>
> [1] Shi, Rongye, Zhaobin Mo, and Xuan Di. "Physics-informed deep learning for traffic state estimation: A hybrid paradigm informed by second-order traffic models." Proceedings of the AAAI Conference on Artificial Intelligence. Vol. 35. No. 1. 2021.
>
> [2] FHWA. Next Generation Simulation (NGSIM) Vehicle Trajectory Data. U.S. Department of Transportation, Federal Highway Administration, 2007.
>
>
> >W2. The paper primarily compares DIF with classical IF and penalty-based IF. It would be beneficial to include comparisons with other state-of-the-art methods for constrained learning or data attribution (e.g., TRAK).
>
> A2. We evaluate counterfactual retraining using three influence estimators: DIF, TracIn [1], and TRAK [2]. For each method, we compute per-sample influence scores on the training set, remove the most harmful 2% and 10% samples, and then retrain the model. We then compare the resulting loss changes on a fixed misclassified test point. The constrained CNN is trained using a primal–dual method.
> For TracIn, we develop a constrained variant following prior work. During primal–dual updates, we record the gradients of the Lagrangian loss with respect to the model parameters. The TracIn score for a training–test pair is computed as $ \mathrm{TracIn}(x_{\mathrm{train}}, x_{\mathrm{test}}) = \sum_{i=1}^{k} \delta_i \, \nabla \ell_{\mathrm{lag}}(\theta_{t_i}, x_{\mathrm{train}}) \cdot\nabla \ell_{\mathrm{lag}}(\theta_{t_i}, x_{\mathrm{test}}), $, where $ \delta_i $ is the step size between checkpoints $ i{-}1 $ and $ i $, and $ \{\theta_{t_1}, \theta_{t_2}, \ldots, \theta_{t_k}\} $ are the parameter checkpoints recorded along the primal–dual trajectory. For TRAK, we apply the method to the penalized objective, where the constraints are incorporated into the loss using a squared hinge penalty.
>
> Both DIF and TracIn achieve substantial loss reductions, with DIF slightly outperforming TracIn, indicating that they can effectively identify the most harmful samples under our constrained training setup. In contrast, TRAK provides little improvement because its random-projection approximation fails to preserve the constraint-adjusted gradient directions, leading to ineffective influence estimates. We have added a detailed discussion of these results in Section 5.2 of the revised manuscript.
>
> Table 1. Loss reduction after removing 2% harmful points (mean ± std).
> | Method | Absolute Loss Reduction |
> |--------|--------------------------|
> | DIF | 0.0143 ± 0.0011 |
> | TracIn | 0.0113 ± 0.0003 |
> | TRAK | 0.0030 ± 0.0002 |
> | Random | 0.0001 ± 0.0011 |
>
> Table 2. Loss reduction after removing 10% harmful points (mean ± std).
> | Method | Absolute Loss Reduction |
> |--------|--------------------------|
> | DIF| 0.0492 ± 0.0008 |
> | TracIn | 0.0452 ± 0.0001 |
> | TRAK | 0.0098 ± 0.0001 |
> | Random | 0.0010 ± 0.0033 |
>
> [1] Pruthi et al. *Estimating training data influence by tracing gradient descent*. NeurIPS 2020.
>
> [2] Park et al. *TRAK: Attributing model behavior at scale*. arXiv:2303.14186.

---

> ### Author Response · Authors · 2025-11-21
> **Response to Reviewer GE37 (Part 2/2)**
>
> > W3. While the theoretical and empirical results are strong, the paper could provide more discussion on the practical implications of DIF, such as computational scalability for large-scale deep learning models.
>
> A3. Thank you for the suggestion. The computational cost of DIF can be analyzed as follows.
>
> DIF computes the update by solving a quadratic program in the variable $ \omega $, which we optimize using a primal–dual method.  The solution $ \omega^\ast $ of this quadratic program converges to the desired parameter update $ \Delta \theta $. Let $ \bar{\theta} $ denote the original model parameters before data perturbation, with parameter dimension $ p $, and let $ N_c $ be the number of active constraints.
>
> At iteration $ t $, updating $\omega$ requires to compute the product between the Hessian at $ \bar{\theta} $ and the current vector $ \omega^{(t)} $, i.e., $\nabla_{\theta \theta}^2 L(\bar{\varepsilon}, \bar{\theta}, \bar{\lambda}) \omega$. Using an Hessian–vector products (HVP), this operation takes $O(p)$ time.
> We also need to compute i) the products between the gradient of the objective at $ \bar{\theta}$ with $\omega$ and dual variables, e.g., $\nabla \theta L\left(x_{\text{train}}, \bar{\theta}\right) \cdot \omega$ and (ii) products between the Jacobian of the constraints at $ \bar{\theta} $, with $ \omega^{(t)} $ and the dual variables, e.g., $\nabla_\theta \ell_j \left(x_{\text{train}}, \bar{\theta}\right) \cdot \omega$. The gradients at $ \bar{\theta} $ can be precomputed and reused. This step costs $O(N_c p)$. Since $ N_c \ll p $, the per-iteration complexity remains $O(p)$.
> Thus, each primal–dual iteration costs $O(p)$. If the primal–dual solver performs $K$ iterations, the overall DIF takes $O(Kp)$ time.
>
> As a comparison, standard influence functions (IF) require $O(p)$ time for a single training sample.
> For large scale models, DIF relies only on Hessian–vector products and Jacobian–vector products, both of which are implemented efficiently in modern autograd systems and scale linearly with the number of parameters.
> We have added a detailed derivation in Appendix F, where we explicitly present the computations in each primal–dual iteration and analyze their corresponding costs.
>
> >W4. The paper briefly mentions the challenges of non-convex optimization in CNNs but does not delve deeply into how DIF handles non-convexity. A more detailed analysis would strengthen the paper.
>
>  A4. Thank you for raising this point. DIF requires some type of local convexity so that Theorem 8 can apply and the QP (18) can be used to calculate the directional derivatives. The SOSC assumption in Theorem 8  specifically requires that, locally at the current solution, the Hessian of the Lagrangian is positive definite over all feasible directions $\Delta \theta$ that do not break the binding constraints. This is the exact mathematical definition of local convexity of the studied optimization model when applying the DIF method.
>
> For CNNs and other non-convex models, DIF constructs a local second-order approximation of the VI system around the current solution  ($\bar{\theta}, \bar{\epsilon}, \bar{\lambda}$) . This approximation ensures that, under small data perturbations, the updated parameters and dual variables remain approximately consistent with the VI conditions. In this way, DIF handles non-convexity by analyzing the local geometry of the solution rather than relying on global convexity assumptions. We have made this clear in Section 4.2 and Section 5.2 of the revised manuscript. One may numerically test if the two assumptions hold for the given solution of a CNN model, which we aim to explore in the revised manuscript or future research.
>
>
>
> >Q. This paper makes a strong contribution to the field of constrained learning by introducing DIF, a theoretically sound and empirically validated method for data attribution. The work is rigorous and addresses a critical gap in existing influence estimation methods. However, the experimental scope and practical implications are weak at this point, therefore I am leaning towards a weak reject.
>
> A. Thank you for your thoughtful assessment. We appreciate that you find the theoretical formulation, empirical validation, and overall rigor of DIF to be strong contributions to constrained learning and influence estimation.
> Regarding the concern about the limited experimental scope and practical implications, we have substantially expanded both aspects in the revised version:
>
> • We added a new constrained application (PINN unlearning) to strengthen the practical relevance of our experiments.
>
>
> • We provided extended ablations comparing DIF, TracIn, and TRAK, with quantitative tables added to Section 5.2.
>
> • We incorporated a detailed analysis of computational complexity in Appendix F.
>
> We hope these additions address your concerns and clarify the broader applicability and practical impact of DIF.

---

### Author Response · Authors · 2025-11-20
**General Response to Reviewers**

We thank the reviewers for their constructive feedback. We are encouraged that all reviewers found the proposed DIF framework a meaningful advance for influence estimation under constraints. We have carefully revised the manuscript to address the raised concerns. Below, we summarize the major updates incorporated in the revised submission:


·       Added comparisons of DIF, TracIn, and TRAK on counterfactual retraining (Section 5.2).

·       Added the performance of IF and penalty-IF in constrained CNNs (Section 5.2).

·       Applied DIF to machine unlearning for Physics-Informed Neural Networks (PINNs), further expanding the empirical scope (Appendix G).

·       Added a detailed analysis of the computational complexity of DIF (Appendix F).

·       Clarified the penalty-IF setup and provided the tuning procedure (Appendix D).

All changes have been added to the manuscript and are highlighted in blue.

---

### Meta-Review · Area_Chair_rK6Y · 2025-12-08

**Summary:**

After reading the manuscript, reviewer comments, and authors response, I summarize my meta-comments as follows.

**Research Question**

This paper considers the data valuation for constrained learning tasks, a new research topic.

**Challenge Analysis**

Influence function is a widely used tool for data valuation. However, when it comes to constrained learning, its estimation becomes inaccurate due to lack of considering the constraints.

**Philosophy**

To consider the constraints, the authors aim to conduct local sensitivity analysis for perturbing data.

**Solution**

The authors cast the optimality conditions as a variational inequality and introduce the Directional Influence Function (DIF) to estimate the change in model solution through a directional derivative approach.

**Theory**

The theoretical part is strong, but can be more informative.

**Experiments**

1. The experiments are not strong, even with the extra PINN experiments. The authors need to demonstrate the effectiveness in practical scenarios to validate DIF for constrained learning.

2. The setting is not clearly illustrated.

3. Only demonstrating the loss reduction is not enough. The authors need to demonstrate the performance improvement in terms of accuracy, reward, robustness, constraint violation that depends on the task. For Figure 5, the authors demonstrate the loss reduction of removing a certain percent of identified detrimental samples. Loss does not directly mean the performance.

4. Figure 3 is good to verify the motivation.

5. The figure visualization can be further improved. Font size is too small.

**Others**

1. The notation can be more informative and consistent.

2. Every equation should ends with a punctuation. For example, Eq. (17).

3. The presentation can be more easy to follow by illustrating the roadmap, rather than directly talking into details. A footnote can be added to show "All proofs are in the appendix," rather than repeating it several times.

**Summary**

I am an expert of the data valuation area and very cheerful to see new papers to thrive in this area. Honestly and frankly, I like the topic. However, the experiments are not strong. I suggest the authors to enrich the experimental part and enhance the presentation for next submission.

**Reviewer Concerns:**

The concerns from reviewers are consensus, which focuses on the experimental part, presentation, non-convex models.

I do not think the authors well addressed the experiments. Significant efforts should be taken to enrich the experimental part.

**Reviewer Scores:**

There is one reviewer who would like to raise the score after the rebuttal.

However, all reviewers and I agree the experimental part is not strong, which makes the current version not meet the bar of ICLR.

---

### Decision · Program_Chairs · 2026-01-26

Reject